# ATP-release pannexin channels are gated by lysophospholipids

Erik Henze[1†], Russell N Burkhardt[2,3], Bennett William Fox[2,3], Tyler J Schwertfeger[2,3], Eric Gelsleichter[3], Kevin Michalski[1], Lydia Kramer[1‡], Margret Lenfest[4], Jordyn M Boesch[4], Hening Lin[3,5,6§], Frank C Schroeder[2,3], Toshimitsu Kawate[1*]

[1]Department of Molecular Medicine, Cornell University, Ithaca, United States; [2]Boyce Thompson Institute, Cornell University, Ithaca, United States; [3]Department of Chemistry and Chemical Biology, Cornell University, Ithaca, United States; [4]Department of Clinical Sciences, College of Veterinary Medicine, Cornell University, Ithaca, United States; [5]Department of Molecular Biology and Genetics, Cornell University, Ithaca, United States; [6]Howard Hughes Medical Institute, Chevy Chase, United States

*For correspondence:
toshi.kawate@cornell.edu

Present address: †Dana-Farber Cancer Institute, Boston, United States; ‡Department of Pharmacology and Molecular Sciences, Johns Hopkins University School of Medicine, Baltimore, United States; §Division of the Biological Sciences, Department of Medicine and Department of Chemistry, University of Chicago, Chicago, United States

Competing interest: The authors declare that no competing interests exist.

## eLife Assessment

Pannexin (Panx) channels are a family of poorly understood large-pore channels that mediate the release of substrates like ATP from cells, yet the physiological stimuli that activate these channels remain poorly understood. The study by Henze et al. describes an elegant approach wherein activity-guided fractionation of mouse liver led to the discovery that lysophospholipids (LPCs) activate Panx1 and Panx2 channels expressed in cells or reconstituted into liposomes. The authors provide **compelling** evidence that LPC-mediated activation of Panx1 is involved in joint pain and that Panx1 channels are required for the established effects of LPC on inflammasome activation in monocytes, suggesting that Panx channels play a role in inflammatory pathways. Overall, this **important** study reports a previously unanticipated mechanism wherein LPCs directly activate Panx channels. The work will be of interest to scientists investigating phospholipids, Panx channels, purinergic signalling and inflammation.
[Editors' note: this paper was reviewed and curated by Biophysics Colab]

**Abstract** In addition to its role as cellular energy currency, adenosine triphosphate (ATP) serves as an extracellular messenger that mediates diverse cell-to-cell communication. Compelling evidence supports that ATP is released from cells through pannexins, a family of membrane proteins that form heptameric large-pore channels. However, the activation mechanisms that trigger ATP release by pannexins remain poorly understood. Here, we discover lysophospholipids as endogenous pannexin activators, using activity-guided fractionation of mouse tissue extracts combined with untargeted metabolomics and electrophysiology. We show that lysophospholipids directly and reversibly activate pannexins in the absence of other proteins. Secretomics experiments reveal that lysophospholipid-activated pannexin 1 leads to the release of not only ATP but also other signaling metabolites, such as 5'-methylthioadenosine, which is important for immunomodulation. We also demonstrate that lysophospholipids activate endogenous pannexin 1 in human monocytes, leading to the release of IL-1β through inflammasome activation. Our results provide a connection between lipid metabolism and purinergic signaling, both of which play major roles in immune responses.

## Introduction

Pannexins, a family of proteins that form heptameric large-pore channels, release signaling molecules like ATP and glutamate from both dying and living cells (*Syrjanen et al., 2021*; *Dahl, 2015*). There are three subtypes (Panx1-3) that share a similar heptameric assembly with a membrane pore capable of allowing such signaling molecules to permeate (*Michalski et al., 2020*; *Ruan et al., 2020*; *Deng et al., 2020*; *Kuzuya et al., 2022*; *Hussain et al., 2024*; *He et al., 2023*; *Zhang et al., 2023*). All three subtypes of pannexins have been demonstrated to play important signaling roles in processes such as immune cell migration and differentiation, epilepsy, migraine, and chronic pain (*Aquilino et al., 2019*; *Muñoz et al., 2021*; *Yeung et al., 2020*; *Harcha et al., 2021*). However, there is limited knowledge about how pannexins are activated, particularly in living cells.

In dying cells, Panx1-mediated ATP-release is important not only for depleting cellular energy and halting metabolism but also for facilitating the recruitment of white blood cells (*Imamura et al., 2020*; *Chekeni et al., 2010*). This so-called 'find-me' signaling allows phagocytes to clear billions of dying cells daily without causing unnecessary inflammation (*Chekeni et al., 2010*). Single-channel recordings and in vitro reconstitution studies demonstrated that cleavage of the C-terminus by caspase during apoptotic cell death triggers Panx1-channel opening (*Narahari et al., 2021*; *Chiu et al., 2017*). The prevailing mechanism involves the C-terminus blocking the pore in its closed conformation, with cleavage of this region facilitating pore unblocking to open the channel (*Ruan et al., 2020*; *Sandilos et al., 2012*). Cleavage of the C-terminus also induces a significant conformational rearrangement in the N-terminus, which appears to play a crucial role in channel gating (*Henze et al., 2024*).

In living cells, other activation stimuli for Panx1 must exist, as the C-terminus remains intact. Studies have suggested that Panx1 is activated through intracellular signaling triggered by the stimulation of structurally unrelated membrane receptors, such as G protein-coupled receptors (e.g. α1-adrenergic receptor)(*Billaud et al., 2011*), ligand-gated ion channels (e.g. NMDA receptor)(*Thompson et al., 2008*), and tumor necrosis factor receptors (*Maier-Begandt et al., 2021*). However, it remains unclear how Panx1 is activated downstream of these apparently unrelated receptors in living cells (*Michalski et al., 2020*; *Ruan et al., 2020*; *Deng et al., 2020*; *Kuzuya et al., 2022*; *Ambrosi et al., 2010*). Furthermore, essentially nothing is known about the activation mechanisms of Panx2 and Panx3.

Our previous studies demonstrated that small molecules, such as probenecid (*Silverman et al., 2008*), could reversibly activate a point mutant of Panx1 (e.g. W74A; *Michalski and Kawate, 2016*). We hypothesized that naturally occurring small molecules could trigger pannexin channel opening. To shed new light on the mechanisms of pannexin activation in living cells, we searched for potential signaling molecules that directly and reversibly activate pannexins.

## Results

### Panx1 and Panx2 are activated by lysophospholipids

To identify potential candidates for endogenous small molecules that regulate pannexin channel opening, we used an activity-guided fractionation approach (*Liu et al., 2018*). We chose mouse liver as our source due to its involvement in diverse metabolic processes and its significantly larger size compared to other organs, making sample collection more feasible. Mouse liver extract was fractionated using reverse-phase chromatography and the fractions were tested for pannexin activation using whole-cell patch-clamp electrophysiology (*Figure 1A*). The soluble fractions were excluded from this study, as the most polar fraction gave rise to strong channel activities in the absence of exogenously expressed pannexins (*Figure 1—figure supplement 1A*). To increase sensitivity of our screen, we employed a Panx1 construct that contains a Gly-Ser insertion at the N-terminus (dubbed 'Panx1+GS'), which yields significantly larger currents than the wildtype channel in HEK293 cells (*Michalski et al., 2018*). For Panx2, we used the wildtype channel, as it gives rise to large currents in HEK293 cells. Panx3 was excluded because it failed to give rise to detectable currents, despite appreciable surface expression in mammalian tissue culture cells (*Figure 1—figure supplement 1B*).

Among the 18 metabolome fractions, two (#11 and #12) gave rise to robust and reversible currents specific to Panx1+GS or Panx2 (*Figure 1—figure supplement 2A, B and F*). Comparative analysis by high performance liquid chromatography coupled to high-resolution mass spectrometry (HPLC-HRMS) revealed approximately 1500 mass spectrometric features that were enriched more than 10-fold in active fractions compared to neighboring fractions that showed negligible activity.

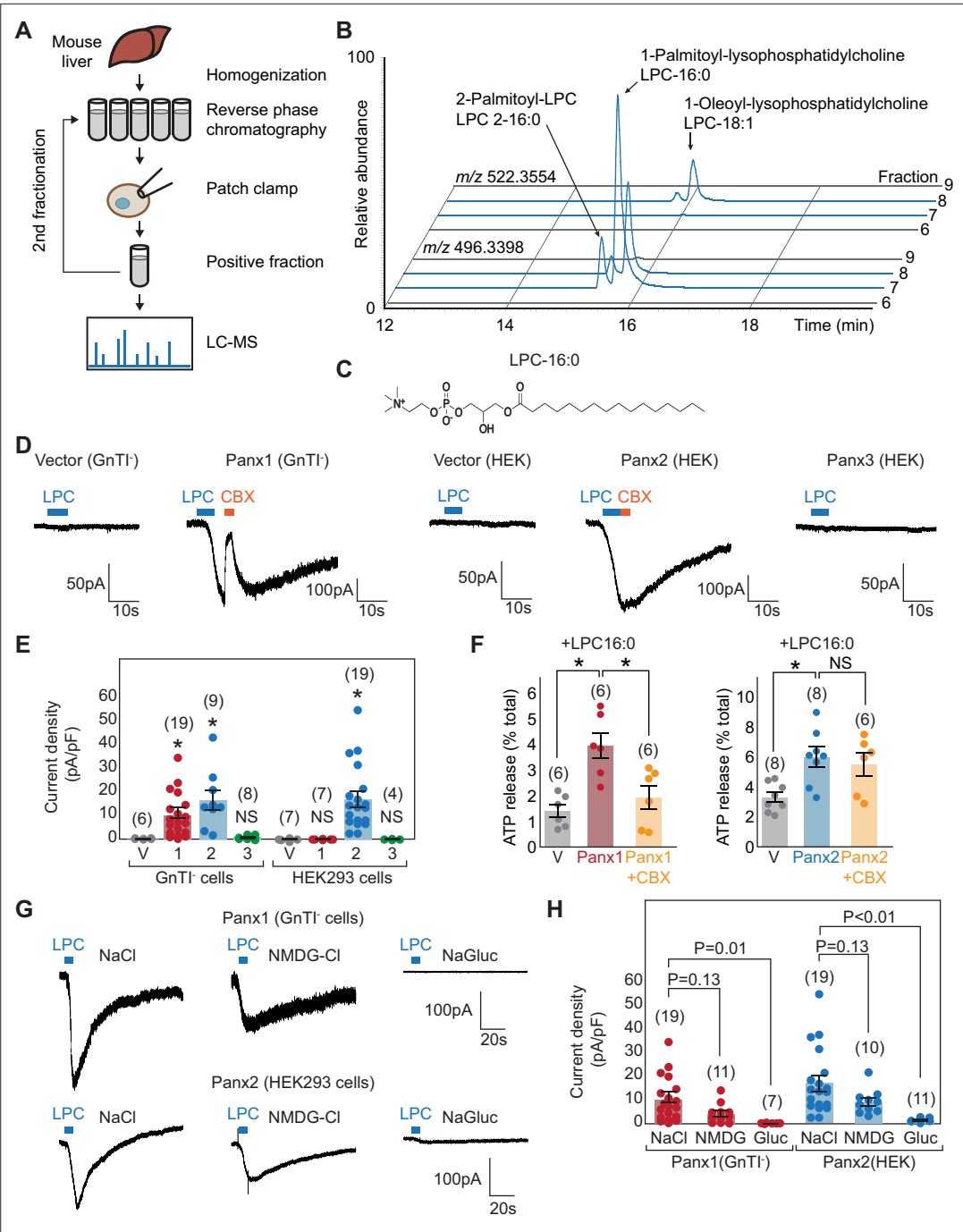

**Figure 1.** Metabolomic screening identifies LPC-16:0 as a pannexin agonist. (**A**) Schematic of the metabolite screen. Organic extracts of mouse liver tissues were fractionated via reverse-phase chromatography and assessed for their ability to stimulate Panx1+GS or Panx2 using whole-cell patch-clamp. Active fractions were analyzed by HPLC-HRMS. (**B**) Extracted ion chromatograms for fractions #5–8 from the second round of Panx1 activity-guided fractionation. (**C**) Chemical structure of LPC-16:0. (**D**) Representative whole-cell patch-clamp traces. (**E**) Quantification of peak current densities triggered by LPC-16:0. Wildtype pannexins were expressed in indicated cells. (**F**) ATP release induced by application of LPC-16:0 to Panx1-expressing GnTI⁻ (left; 10 µM at 3 min) and Panx2-expressing HEK293 (right; 30 µM at 3 min) cells. Data are expressed as percent of total ATP released upon membrane solubilization. P values were calculated using unpaired t-test with unequal variances. An asterisk denotes p<0.01. Error bars represent SEM. (**G**) Representative whole cell currents of Panx1 and Panx2 stimulated by LPC-16:0 in buffers containing different anions and cations. (**H**) Quantification of peak current densities in different buffers. Voltage-clamp recordings were performed at –60 mV. Blue bars indicate application of LPC-16:0

*Figure 1 continued on next page*

*Figure 1 continued*

(7 µM), and orange bars indicate application of carbenoxolone (50 µM). The number of recordings from different cells for each experiment (biological replicates) is shown in parenthesis. One-way ANOVA followed by Dunnett's test was used to assess statistical significance. V indicates the vector control.

The online version of this article includes the following source data and figure supplement(s) for figure 1:

**Source data 1.** Electrophysiology and ATP release assay data used for *Figure 1*.

**Figure supplement 1.** Characteristics of the pannexins and other large pore forming channels used in this study.

**Figure supplement 1—source data 1.** Electrophysiology data used for *Figure 1—figure supplement 1*.

**Figure supplement 1—source data 2.** Raw blot images used for *Figure 1—figure supplement 1*.

**Figure supplement 1—source data 3.** Annotated blot images used for *Figure 1—figure supplement 1*.

**Figure supplement 2.** Representative whole-cell patch-clamp traces for mouse liver fractions.

**Figure supplement 2—source data 1.** Electrophysiology data used for *Figure 1—figure supplement 2*.

To narrow down the candidate metabolites, we pooled the active fractions and performed a second round of activity-guided fractionation. Two of the resulting fractions (#7 and #8) strongly activated both Panx1+GS and Panx2 (*Figure 1—figure supplement 2C–E and G*); comparative metabolomic analysis revealed twelve major metabolites at least 10-fold enriched in the active fractions relative to the neighboring inactive fractions (i.e., #6 and #9). Analysis of their MS/MS fragmentation spectra and comparison with authentic standards indicated that the majority of the differential metabolites were lysolipids, including both isomers of palmitoyl-lysophosphatidylcholine (LPC-16:0), palmitoyl-lysophosphatidylethanolamine (LPE-16:0), and oleoyl-LPC (LPC-18:1), as well as an additional poly-unsaturated LPC-20:3 (*Figure 1B*). LPCs are known to serve diverse signaling roles, especially in inflammation (*Law et al., 2019*), and exist in extracellular fluids at high micromolar concentrations (*Psychogios et al., 2011*), similar to the concentrations of these metabolites in the active fractions.

To test whether LPCs activate pannexins, we performed whole-cell patch-clamp recordings for the wildtype pannexin channels using synthetic compounds. We focused on LPC-16:0 (*Figure 1C*), which is commercially available and the most abundant LPC variant in extracellular fluids (*Tan et al., 2020*; *Ojala et al., 2007*). For Panx1, we used HEK293S GnTI⁻ cells, an HEK derivative commonly used for structural studies (*Reeves et al., 2002*), as we observed significantly stronger currents in these cells. As for Panx2, we continued using HEK293 cells, as they exhibited lower background signals. Cell-surface-expression analysis revealed that both Panx1 and Panx2 were expressed at higher levels in GnTI⁻ cells than in HEK293 cells, potentially contributing to both the increased sensitivity and background signal (*Figure 1—figure supplement 1B*). A robust and reversible current was observed for both Panx1 and Panx2 following stimulation with LPC-16:0 (*Figure 1D*), while Panx3 and other large-pore channels—such as LRRC8A, connexin 43, and innexin 6—failed to show measurable currents under the same experimental conditions (*Figure 1—figure supplement 1C and D*). Panx3 current was not observed in either HEK293 or GnTI⁻ cells, despite detectable surface expression (*Figure 1E*, *Figure 1—figure supplement 1B and D*). Recordings from vector-transfected cells and carbenoxolone (CBX) sensitivity confirmed that the LPC-16:0-mediated currents were specific to pannexins; Panx2 was insensitive to CBX (*Figure 1E*), consistent with previous findings (*Poon et al., 2014*). Importantly, extracellular ATP levels significantly increased when Panx1- or Panx2-expressing cells were stimulated with LPC-16:0 (*Figure 1F*). These results indicate that LPCs may function as endogenous signaling molecules to promote ATP-release from living cells through activation of Panx1 and/or Panx2.

To investigate the ion selectivity of lysophospholipid-activated Panx1 and Panx2 channels, we compared whole-cell current density before and after LPC treatment at –60 mV using buffers containing various anions and cations. Both Panx1 and Panx2 channels exhibited significantly larger currents in NaCl or N-Methyl-D-glucamine chloride (NMDG-Cl) compared to sodium gluconate (NaGluc), indicating that lysophospholipid-activated channels are more selective for anions under these conditions (*Figure 1G and H*). This finding aligns with the reported ion selectivity of voltage-stimulated Panx1 channels (*Romanov et al., 2012*; *Ma et al., 2012*). Interestingly, currents in NMDG-Cl were slightly smaller than in NaCl, suggesting that NMDG may have an inhibitory effect on LPC-16:0-activated channels. While we acknowledge that this analysis does not directly compare ion selectivity within the same patch, the nearly negligible current observed in NaGluc strongly indicates that the anion conductance through both Panx1 and Panx2 channels is greater than cation conductance.

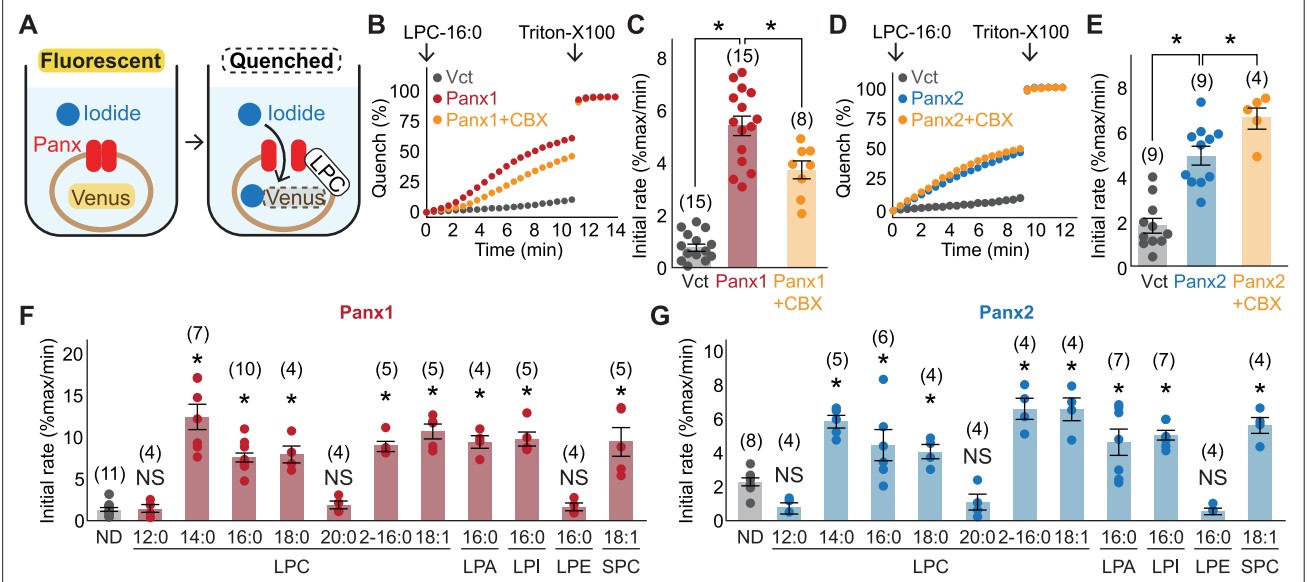

**Figure 2.** Cellular mVenus-quench assays reveal a series of lysophospholipids as pannexin agonists. (**A**) Cartoon illustrating the principle of the mVenus quench assay. (**B–E**) Representative traces (**B**) for Panx1 and (**D**) for Panx2 and quantification of initial mVenus quenching rates (**C**) for Panx1 and (**E**) for Panx2. LPC-16:0 (30 µM) was applied with or without CBX (50 µM), and the maximum mVenus quenching was measured after cell solubilization with 1% Triton-X100. P values were calculated using unpaired t-test with unequal variances. (**F**) and (**G**) Initial mVenus quenching rates of Panx1 expressed in GnTI⁻ cells (**F**) and Panx2 expressed in HEK293 cells (**G**). Pannexin activation was measured following addition of 60 µM sn-1 LPCs (LPC-12:0-20:0), sn-2 LPC (LPC2-16:0), a monounsaturated sn-1 (LPC-18:1), or other sn-1 lysophospholipids with different headgroups (LPA-16:0, LPI-16:0, LPE-16:0, and SPC-18:1). The number of measurements from different cells for each experiment (biological replicates) is shown in parenthesis. P values were calculated using one-way ANOVA followed by Dunnett's t-test. Asterisks denote p<0.01. Error bars represent SEM.

The online version of this article includes the following source data and figure supplement(s) for figure 2:

**Source data 1.** mVenus quench assay data used for *Figure 2*.

**Figure supplement 1.** Lysophospholipid dose-response curves obtained from the mVenus quench assay.

**Figure supplement 1—source data 1.** mVenus quench assay data used for *Figure 2—figure supplement 1*.

We next tested structurally diverse lysophospholipids for pannexin activity. To facilitate screening, we developed a fluorescence-based assay in which iodide influx through pannexin channel is measured by fluorescence-quenching of a halide biosensor mVenus (*Figure 2A*; *Nagai et al., 2002*; *Galietta et al., 2001*). This pannexin-mediated decrease in signal is normalized to the maximal quenching obtained via membrane permeabilization with the detergent, Triton-X 100. Addition of LPC-16:0 robustly quenched mVenus fluorescence in cells expressing Panx1 or Panx2 (*Figure 2B–E*). Likewise, LPCs with 14:0, 18:0, 2-16:0, or 18:1 (oleoyl) acyl groups activated both Panx1 and Panx2 with EC50 values within the 10–50 µM range (*Figure 2F, G*, *Figure 2—figure supplement 1*). In contrast, neither LPC-12:0 nor LPC-20:0 activated these pannexins, suggesting that the most effective LPC acyl chain length is between 14 and 18 carbons. Lysophospholipids with other headgroups, including lysophosphatidic acid (LPA), lysophosphatidylinositol (LPI), and sphingosylphosphorylcholine (SPC) were as potent as LPC, whereas lysophosphatidylethanolamine (LPE) failed to activate pannexins even at much higher concentration (*Figure 2F, G*, *Figure 2—figure supplement 1*). Together, these results suggest that pannexins are activated by select lysophospholipids; considering that the extracellular concentrations of LPCs are orders of magnitude higher than the other tested species (*Tan et al., 2020*), LPCs are the most likely activators of Panx1 and Panx2 in vivo.

## LPC-16:0 stimulation of Panx1 triggers the release of immunomodulatory metabolites

In addition to ATP, Panx1 has been demonstrated to release metabolites that play crucial roles in the resolution of inflammation and the maintenance of effector T cell populations post-infection (*Medina et al., 2020*; *Vardam-Kaur et al., 2024*). We conducted comparative metabolomics to investigate

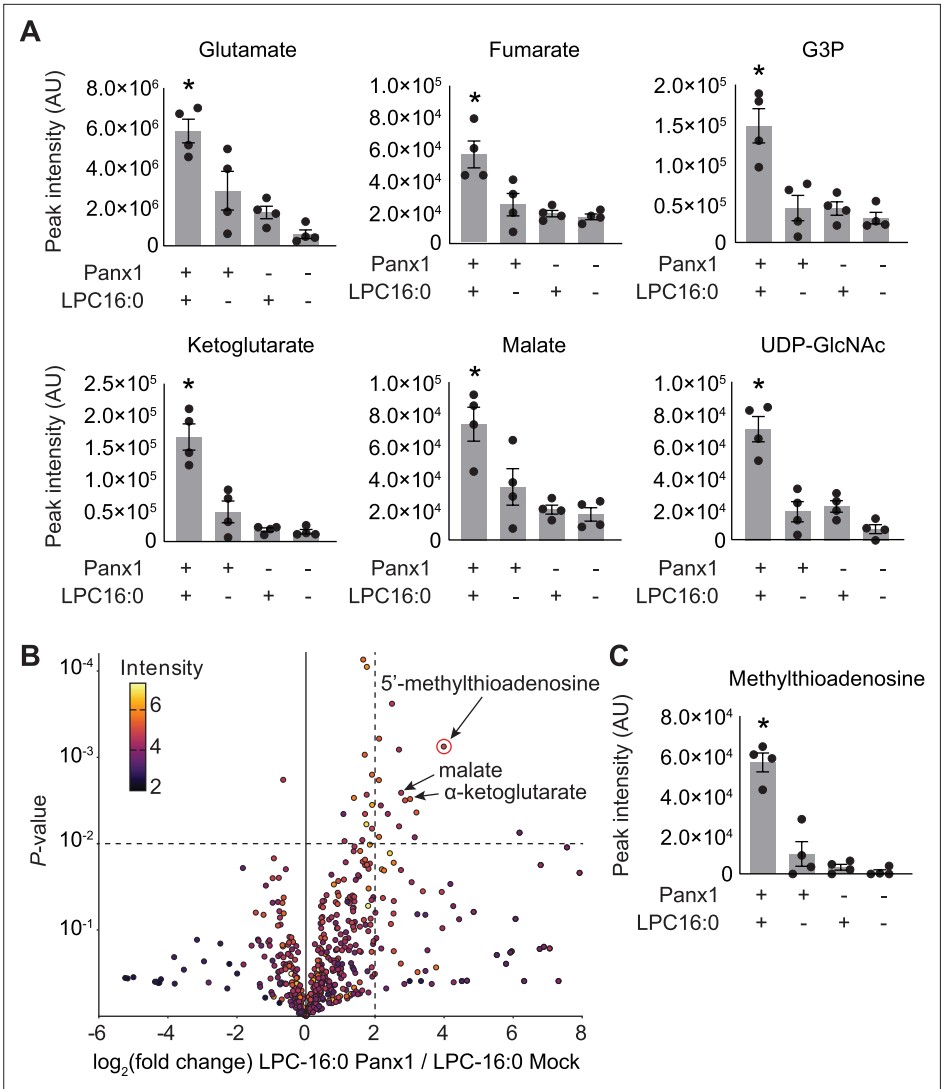

**Figure 3.** Select released signaling metabolites following LPC-16:0 stimulation of Panx1. **A** Metabolites enriched in the conditioned media (CM) of cells expressing Panx1 treated with LPC-16:0 that were previously identified as Panx1 permeant using apoptotic T cells (**Medina et al., 2020**). (**B**) Comparative analysis by HPLC-HRMS of CM from cells expressing Panx1 +LPC-16:0 versus CM from vector-expressing cells treated with LPC-16:0. Volcano plot depicts subset of features detected in negative ion mode. Unadjusted p-values calculated by unpaired, two-sided *t*-test (see Materials and methods for details). (**C**) Additional metabolite discovered in this study with known roles in immunomodulation. Panx1 was expressed in GnTI⁻ cells and the released metabolites were analyzed 45 min after the stimulation with LPC-16:0 (10 µM). N=4. p values were calculated using one-way ANOVA. Asterisks denote p<0.05. Error bars represent SEM.

The online version of this article includes the following source data for figure 3:

**Source data 1.** Mass spec data used for *Figure 3*.

whether lysophospholipids trigger the release of these or other metabolites through Panx1 channels. GnTI⁻ cells expressing empty vector or Panx1 were treated with vehicle or LPC-16:0, and the metabolites in the conditioned media were analyzed by HPLC-HRMS. Notably, 6 of the 25 metabolites previously identified as Panx1 permeants from apoptotic T cells (**Medina et al., 2020**) were enriched in the medium of Panx1-expressing cells stimulated with LPC-16:0 (*Figure 3A*). These findings indicate that the selectivity of the Panx1 channel opened by lysophospholipids is comparable to the selectivity of the channel when opened by caspase-dependent C-terminal cleavage. Untargeted comparative analysis revealed additional enriched metabolites released from LPC-16:0-stimulated Panx1-expressing cells, including the immunomodulatory metabolite 5'-methylthioadenosine (*Figure 3B and C*; *Suzzi*

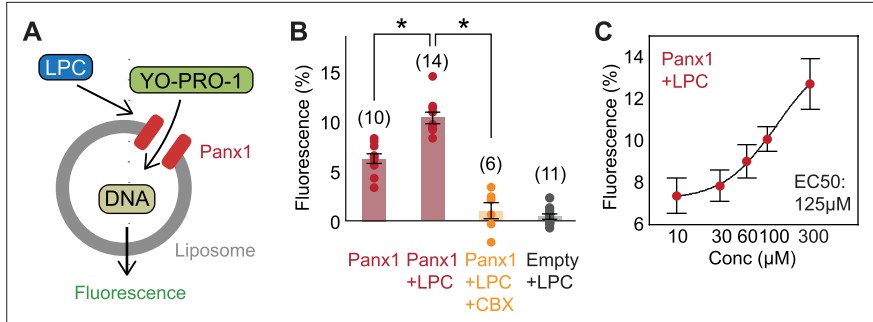

**Figure 4.** Functional reconstitution of Panx1 confirms direct activation by LPC-16:0. **A** Schematic representation of YO-PRO-1 uptake assay. (**B**) Relative YO-PRO-1 fluorescence triggered by LPC-16:0 (100 μM) with or without CBX (50 μM). Asterisks indicate p<0.01 using unpaired t-test. The number of measurements from different cells for each experiment (biological replicates) is shown in parenthesis. (**C**) Dose-response profile of Panx1 treated with LPC-16:0. Dose responses were fitted with the Hill equation, and the EC50 values are indicated. The number of biological replicates is available in *Figure 4—source data 1*. Error bars represent SEM.

The online version of this article includes the following source data and figure supplement(s) for figure 4:

**Source data 1.** YO-PRO-1 uptake assay data used for *Figure 4*.

**Figure supplement 1.** Functional reconstitution of purified Panx1.

**Figure supplement 1—source data 1.** YO-PRO-1 uptake assay data used for *Figure 4—figure supplement 1*.

**Figure supplement 1—source data 2.** Raw gel images used for *Figure 4—figure supplement 1*.

**Figure supplement 1—source data 3.** Annotated gel images used for *Figure 4—figure supplement 1*.

*et al., 2023*). These data support the idea that lysophospholipids modulate inflammatory response by stimulating Panx1-dependent release of signaling metabolites beyond ATP.

## Lysophospholipids directly activate Panx1

Evidence for several lysophospholipid-activated channels and membrane receptors (*Tan et al., 2020*; *Marra et al., 2016*; *Maingret et al., 2000*) raises the possibility that pannexin agonism by LPC may be indirect. To test whether lysophospholipids directly activate pannexins, we performed a functional reconstitution in vitro to investigate pannexin activity in the absence of other proteins (*Figure 4A*). The full-length Panx1 was purified from GnTI⁻ cells using affinity and size exclusion chromatography. Purified Panx1 (*Figure 4—figure supplement 1A and B*) was reconstituted into proteoliposomes composed of 1-palmitoyl-2-oleoyl-sn-glycero-3-phosphoethanolamine (POPE), 1-palmitoyl-2-oleoyl-sn-glycero-3-phosphoglycerol (POPG), and sphingomyelin (SM; brain extract). We assessed Panx1 channel activity through YO-PRO-1 dye uptake, a well-established method for demonstrating the activity of large-pore-forming channels using proteoliposomes (*Karasawa et al., 2017*; *Pelegrin and Surprenant, 2006*). We also confirmed that LPC-16:0 triggers YO-PRO-1 uptake in Panx1-expressing GnTI⁻ cells using a cell-based assay (*Figure 4—figure supplement 1C and D*). Upon LPC-16:0 application, Panx1-reconstituted proteoliposomes took up YO-PRO-1 in a dose-dependent manner (*Figure 4B and C*). Control proteoliposomes lacking Panx1 or CBX-treated Panx1-liposomes confirmed that the observed YO-PRO-1 uptake was Panx1 dependent. The direct effect of lysophospholipids on this channel is likely conserved across species, as we also observed LPC-16:0-dependent YO-PRO-1 uptake using proteoliposomes reconstituted with the frog Panx1 construct previously employed for cryo-EM studies (*Michalski et al., 2020*; *Figure 4—figure supplement 1E–H*). These results provide evidence that lysophospholipids directly activate Panx1 in the absence of other proteins.

## Phospholipase A mediates pannexin activation

Lysophospholipids are produced from membrane phospholipids primarily by phospholipase A1 (PLA1) or phospholipase A2 (PLA2) enzymes (*Tan et al., 2020*), which hydrolyze phospholipids to produce lysophospholipids and free fatty acids important in inflammation (*Vasquez et al., 2018*; *Figure 5A*). Considering that extracellular ATP concentration increases during inflammation (*Dosch et al., 2018*), we wondered whether phospholipase A activation may lead to increased abundance of

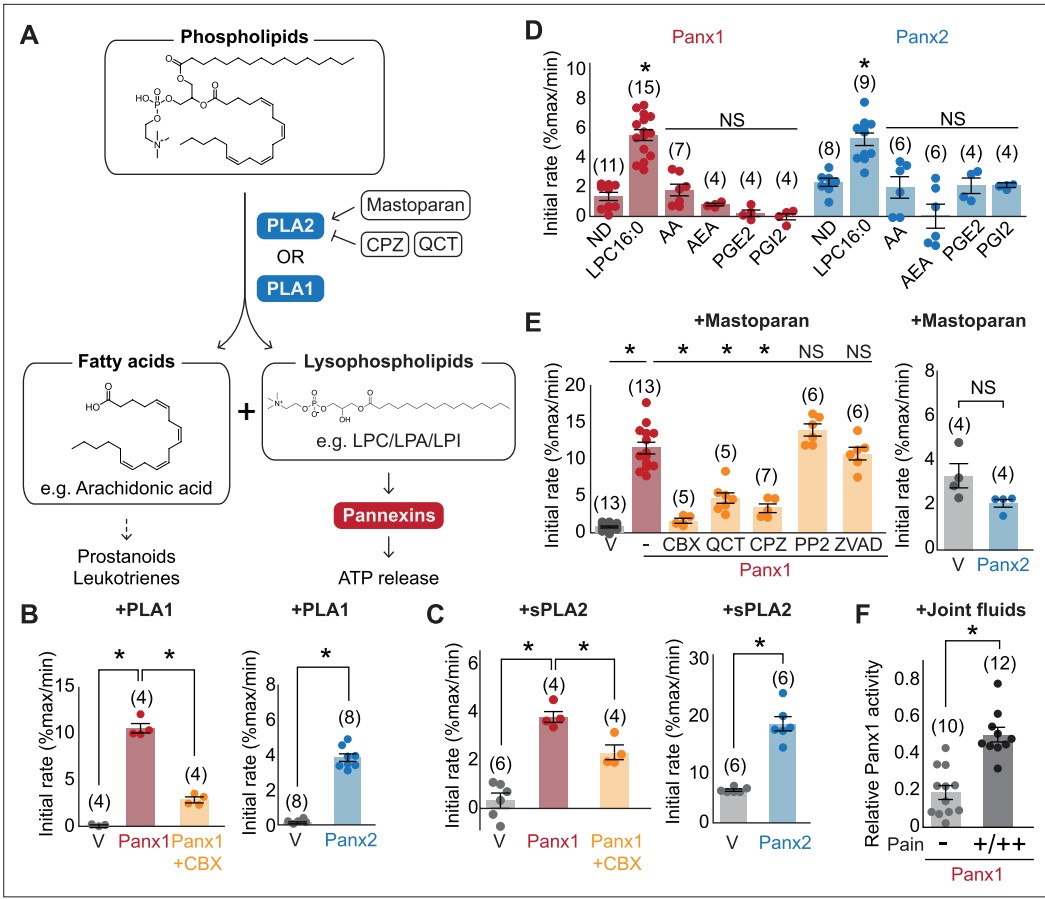

**Figure 5.** Pannexins mediate lysophospholipid signaling. (**A**) Schematic illustrating lysophospholipid signaling. (**B–E**) Pannexin activation triggered by extracellularly applied stimuli. Normalized initial mVenus quenching rates are shown for PLA1 (**B**), sPLA2 (**C**), and major metabolic products of PLA2 (**D**), mastoparan with or without PLA2 inhibitors (QCT and CPZ), a Src kinase inhibitor (PP2), or a caspase inhibitor (ZVAD) (**E**). V indicates the vector control. Panx1 was expressed in GnTI⁻ cells and Panx2 was expressed in HEK293 cells. (**F**) Panx1-dependent mVenus quenching induced by synovial fluids obtained from canine patients with mild (-) or moderate/severe (+/++) pain. The activity of each fraction was normalized to the effect of LPC-16:0 (30 µM). Each point represents a different patient. P values were calculated using unpaired Student's t-test with unequal variances (**B**), (**C**), and (**F**) or using one-way ANOVA, followed by Dunnett's t-test (**D and E**). The number of measurements from different cells for each experiment (biological replicates) is shown in parenthesis. Asterisks indicate p<0.01. Error bars represent SEM.

The online version of this article includes the following source data for figure 5:

**Source data 1.** mVenus quench assay data used for *Figure 5*.

lysophospholipids and therefore pannexin channel opening. To test this possibility, we first assessed whether extracellular application of PLA1 or sPLA2, a secreted form of this enzyme, can open pannexin channels. As with LPC-16:0, we found that application of PLA1 or sPLA2 resulted in robust mVenus quenching for both Panx1- and Panx2-expressing cells (*Figure 5B and C*). Since PLA2 also leads to the production of signaling molecules other than lysophospholipids, we tested some of these representative lipid metabolites known to mediate inflammatory responses (*Dennis and Norris, 2015*). However, none of the tested lipid species (i.e., arachidonic acid, N-arachidonoylethanolamine, prostaglandin E2, or prostaglandin I2) triggered mVenus quenching (*Figure 5D*). These results suggest that lysophospholipids, but not free fatty acids or other lipid mediators, generated from the plasma membrane can activate pannexins.

We next tested whether endogenously existing cytoplasmic PLA2 (e.g., cPLA2) can activate pannexins. PLA2 is activated by polycationic amphipathic peptides commonly found in the venom of poisonous creatures. One such peptide is mastoparan, a toxic component of wasp venom that can

induce cPLA2 activity in a variety of cell types (*Joyce-Brady et al., 1991*; *Gil et al., 1991*). Indeed, application of mastoparan caused a robust, CBX-sensitive mVenus quenching in cells expressing Panx1 (*Figure 5E*). This quenching was attenuated by the PLA2 inhibitors chlorpromazine (CPZ) and quercetin (QCT). Because mastoparan application may trigger Panx1 activation through other mechanisms, such as C-terminal cleavage by caspase or activation of Src kinases (*Lohman et al., 2019*), we applied mastoparan following preincubation with a pan-caspase inhibitor Z-DEVD-FMK or the Src inhibitor PP2. However, neither inhibitor affected mastoparan-dependent mVenus quenching, indicating that indirect modes of Panx1 activation were unlikely (*Figure 5E*). Interestingly, mastoparan-mediated mVenus quenching was not observed in cells expressing Panx2. It is possible that Panx2 may prefer lysophospholipids produced from the outer membrane leaflet. Together, these experiments indicate that both PLA1 and PLA2 can activate pannexins, likely through the production of lysophospholipids from the plasma membrane.

## Synovial fluid from canine patients experiencing pain stimulates Panx1

Given that lysophospholipid and phospholipase concentrations are elevated in patients suffering from joint diseases (*Jacquot et al., 2022*), we hypothesized that synovial fluids from dogs with naturally occurring algogenic disease might trigger Panx1 activation. To test this hypothesis, we investigated Panx1 channel activation using joint fluid collected from 22 canine patients suffering from varying degrees of pain. We found that the joint fluid collected from dogs with moderate to severe pain triggered a robust mVenus quenching (*Figure 5F*). In contrast, joint fluid collected from dogs assessed to have only mild pain showed significantly weaker mVenus quenching (*Figure 5F*). These data demonstrate a correlation between pain-related behaviors and naturally occurring metabolites in synovial joint fluid that can activate Panx1.

## LPC-16:0 activates endogenous Panx1 and induces the release of IL-1β from monocytes

Lysophospholipids have been demonstrated to activate the inflammasome in various cell types, including endothelial cells, microglia, and monocytes, exemplified by the cleavage and release of interlukin-1β (IL-1β); (*Scholz and Eder, 2017*; *Freeman et al., 2017*; *Corrêa et al., 2019*; *Rimola et al., 2020*). A recent study suggested that this mechanism relies on the P2X7 receptors, which are activated by extracellular ATP (*Ismaeel and Qadri, 2021*). Interestingly, LPC-dependent ATP and IL-1β release were blocked by CBX or probenecid, suggesting that the release of these signaling molecules may be mediated by Panx1.

To clarify whether lysophospholipids activate endogenous Panx1 to trigger inflammasome formation, we assessed IL-1β release from phorbol 12-myristate 13-acetate (PMA)-differentiated human THP-1 monocytes. Western blot analysis confirmed the expression of endogenous Panx1 protein in PMA-differentiated THP-1 cells, and two independent shRNAs effectively reduced its expression (*Figure 6A*). While lipopolysaccharide (LPS) treatment alone promoted inflammasome activation to some extent, LPC-16:0 treatment increased the amount of released cleaved IL-1β by approximately 10 fold (*Figure 6B*). Panx1 cleavage was minimal and similar between the LPS-only and LPS +LPC-16:0 treated control cells, and control cells treated with LPC-16:0 alone did not promote Panx1 cleavage, indicating that the increase in IL-1β release with LPC-16:0 treatment is not due to further LPC-induced cleavage of Panx1 (*Figure 6A*). Conversely, Panx1-knockdown cells released less cleaved IL-1β into the supernatant and accumulated intracellular pro-IL-1β (*Figure 6B and C*). The expression levels of NLRP3 and pro-caspase 1 remained comparable among all samples, eliminating the possibility that differences in the expression levels of these key inflammasome components might explain any differences in the maturation and release of IL-1β (*Figure 6—figure supplement 1*). These findings suggest that LPC stimulation of endogenous Panx1 contributes to inflammasome activation in human monocytes.

## Discussion

In this study, we demonstrate that both Panx1 and Panx2 are directly and reversibly activated by lysophospholipids at naturally occurring concentrations in body fluids produced from plasma phospholipids by PLA enzymes (*Figure 7*). This bona fide stimulus triggers the release of signaling molecules

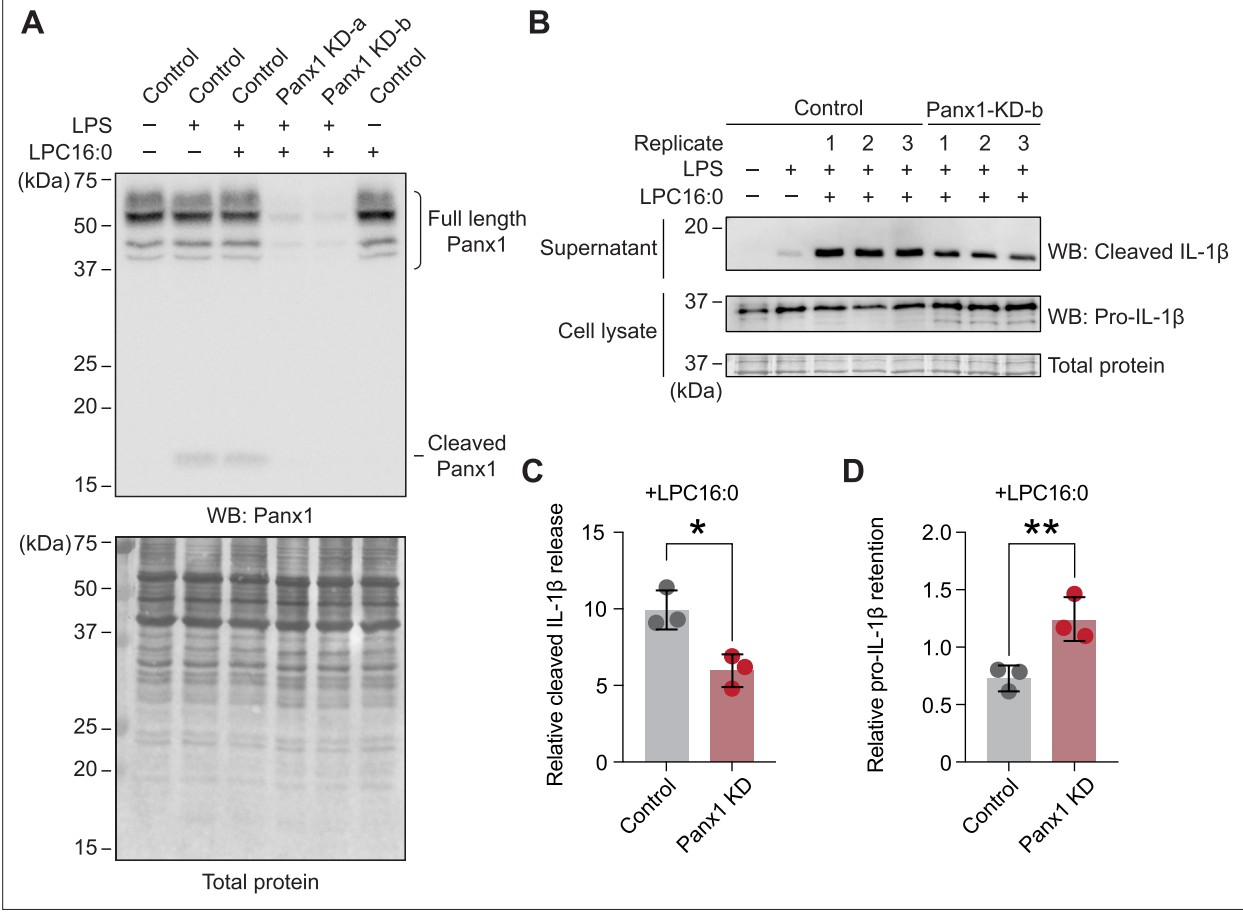

**Figure 6.** Knockdown of endogenous Panx1 reduces LPC-16:0-triggered release of cleaved IL-1β. (**A**) Panx1 protein expression levels in PMA-differentiated/LPS-primed THP-1 control (shRNA-empty vector) cells or two different *shPANX1* knockdown (KD) lines. (**B**) Cleaved IL-1β released into culture supernatant following stimulation with 50 µM LPC-16:0 for 1.5 hr. A representative blot for control and Panx1 knockdown cells is shown. (**C**) Densitometry (ImageJ) was used to quantify the LPC-16:0-induced release of cleaved IL-1β from control and Panx1 KD-b cells relative to the release from control cells only primed with LPS. (**D**) Relative amount of the retained pro-IL-1β normalized to total protein in the cells. p values were calculated using unpaired Student's t-test. * indicates p<0.0032 and ** indicates p<0.0021. Error bars represent SD. Three individual experiments using different batches of cells (biological replicates) were performed.

The online version of this article includes the following source data and figure supplement(s) for figure 6:

**Source data 1.** Densitometry data used for *Figure 6*.

**Source data 2.** Raw blot images used for *Figure 6*.

**Source data 3.** Annotated blot images used for *Figure 6*.

**Figure supplement 1.** Expression of Panx1 and other inflammasome components in THP-1 cells treated with LPC16:0.

**Figure supplement 1—source data 1.** Raw blot images used for *Figure 6—figure supplement 1*.

**Figure supplement 1—source data 2.** Annotated blot images used for *Figure 6—figure supplement 1*.

including ATP and the newly characterized Panx1 permeant MTA from living cells, uniting lipid and purinergic signaling pathways important for inflammation. Our in vitro reconstitution experiments demonstrate that full-length Panx1 can be activated by lysophospholipids in the absence of other proteins, providing compelling evidence that this channel possesses an intrinsic mechanism for activation in this manner. Considering that lysophospholipid concentrations are elevated in many inflammatory diseases (*Tan et al., 2020*; *Kano et al., 2022*), pannexin-mediated ATP release likely plays a critical role in such pathological conditions. This is consistent with our experiments demonstrating that joint fluid from dogs experiencing pain triggers robust Panx1 activation.

The discovery of lysophospholipids as a common stimulus for Panx1 and Panx2 may underlie the proposed compensatory roles between Panx1 and Panx2 described in ischemic stroke (*Bargiotas*

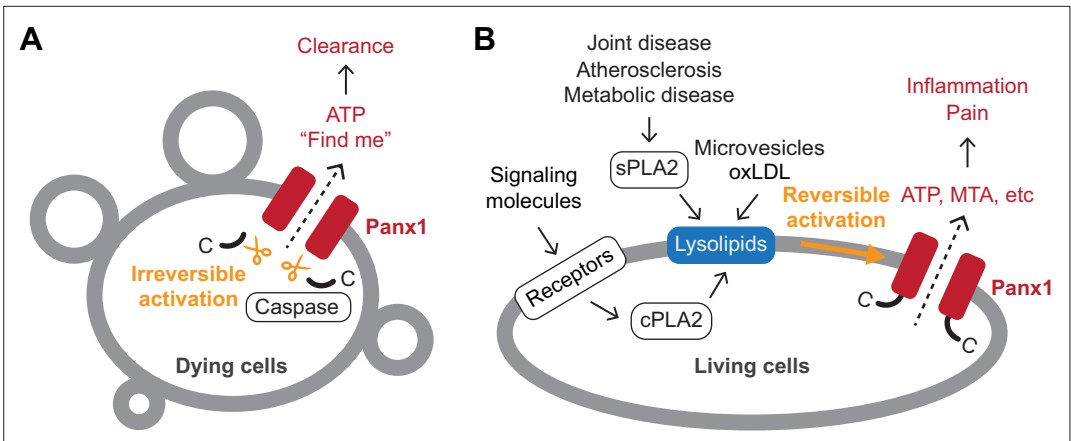

**Figure 7.** Schematic summary of Panx1-mediated signaling. **A** Panx1 activation in dying cells. In dying cells, the C-terminal tails of Panx1 are cleaved by caspases, initiating an irreversible mode of Panx1 activation. This leads to the release of 'find-me' signals, such as ATP, which play a role in attracting phagocytic cells to the site of cell death. (**B**) Panx1 activation in living cells. In living cells, activation of membrane receptors such as NMDA, P2X7, TNF-α, and α1-adrenergic receptors stimulates the production of lysophospholipids via cytoplasmic phospholipase A (PLA) enzymes. Lysophospholipids are abundant in extracellular microvesicles and oxidized low-density lipoproteins (oxLDLs). They are also produced by secreted PLAs during pathological conditions, including atherosclerosis and joint or metabolic diseases. These lysophospholipids reversibly activate Panx1, leading to the release of signaling molecules crucial for inflammation and pain.

*et al., 2011*) and insulin secretion from β-cells (*Berchtold et al., 2017*). Given that Panx3 is ~45% identical to Panx1 and shares a similar overall structure (*Hussain et al., 2024*), it is surprising that Panx3 was not activated by lysophospholipids. Since Panx3 does not generate any currents in our heterologous expression systems, it is possible that Panx3 requires a currently uncharacterized post-translational modifications or binding partners for activation by lysolipids. Alternatively, it may respond to a different class of metabolites.

Lysophospholipid-mediated pannexin activation makes biological sense for several reasons. First, PLA enzymes are activated downstream of NMDA, P2X7, TNF-α, and α1-adrenergic receptors, which have been demonstrated to lead to pannexin channel opening following their stimulation (*Billaud et al., 2011*; *Thompson et al., 2008*; *Maier-Begandt et al., 2021*; *Pelegrin and Surpranant, 2006*; *Burch et al., 1986*; *Alzola et al., 1998*; *Tapia-Arancibia et al., 1992*; *Hoeck et al., 1993*). Second, the concentrations of lysophospholipids increase under pathological conditions, in which pannexins play significant roles. For example, vascular inflammation caused by platelet-derived microvesicles can be explained by the action of extracellular ATP released through Panx1 channels, as these microvesicles contain a large amount of LPCs (*Diehl et al., 2019*). Likewise, potentiation of angiotensin-II dependent vasoconstriction by oxidized low-density lipoproteins (oxLDLs)—a condition associated with vasospasm in atherosclerotic arteries—may be mediated by Panx1 activation, since oxLDLs are rich in LPCs (*Galle et al., 2003*). Third, both augmented Panx1 channel expression and elevated levels of lysophospholipids have been independently reported to be associated with insulin resistance (*Law et al., 2019*; *Tozzi et al., 2020*), a common health problem linked to a wide array of pathologic conditions, including type 2 diabetes, hypertension, and atherosclerosis.

We also demonstrated that lysophospholipids can induce the release of signaling metabolites beyond ATP through Panx1 channel activation. While it is possible that this metabolite release occurs indirectly, the overlap of several permeants with previous studies involving apoptotic T cells (*Medina et al., 2020*) suggests that the Panx1 channel activated by lysophospholipids in living cells likely exhibits a similar selectivity to that observed following C-terminal cleavage in dying cells. The finding of MTA as a novel Panx1-secreted metabolite opens a new area of inquiry into the role of this membrane channel in immune cell signaling, particularly as it relates to cancer (*Li et al., 2019*). MTA acts as an adenosine receptor agonist to tamp down the inflammatory response, which discourages infiltration of T cells and NK cells to the tumor microenvironment (*Jacobs et al., 2020*). Given the

emerging positive link between Panx1 expression and various cancers (*Laird and Penuela, 2021*), it is possible that this metabolite is responsible for this trend.

Our findings suggest that lysophospholipids activate endogenous Panx1 in human THP-1 cells, leading to inflammasome activation. This is consistent with a previous study demonstrating that lysophospholipids stimulate ATP release and inflammasome activation (*Ismaeel and Qadri, 2021*). Although previous studies using bone marrow derived macrophages from Panx1-knockout mice suggested that this channel is dispensable for P2X7-dependent inflammasome activation (*Qu et al., 2011*), those experiments utilized both nigericin (an ionophore known to activate potassium efflux and the inflammasome) and exogenous ATP, which would directly activate the P2X7 receptor without the upstream endogenous ATP-release event. Indeed, Ismaeel et al. showed that IL-1β release was blocked by either apyrase or P2X7 receptor antagonists (*Ismaeel and Qadri, 2021*). Furthermore, Yang et al. showed that Panx1 was required for IL-1β release and with P2X7 exacerbated mortality in a mouse model of sepsis (*Yang et al., 2015*). Interestingly, our experiments revealed that LPS stimulation alone triggered some cleavage of Panx1 and release of cleaved IL-1β to a limited extent. The limited liberation of mature IL-1β from control cells treated with LPS alone may be attributable to the Panx1-independent action of LPS on large-conductance potassium channels (*Parzych et al., 2017*). Nevertheless, although the underlying mechanism of Panx1 cleavage remains unclear, this cleavage may contribute to IL-1β release from cells treated only with LPS. Additionally, we observed attenuated IL-1β release in Panx1-knockdown cells; residual IL-1β release may be attributed to incomplete suppression of gene expression or through the opening of two-pore domain potassium (K2P) channels known to be involved in inflammasome activation (*Madry et al., 2018*; *Di et al., 2018*; *Immanuel et al., 2024*) and activated by lysophospholipids (*Maingret et al., 2000*).

Our functional reconstitution studies revealed some basal activity of the full-length Panx1 channel, which is inconsistent with findings from the previous study (*Narahari et al., 2021*). While the underlying mechanism remains unclear, there are substantial differences between the two experimental setups that may account for the divergent results. First, we used human Panx1 tagged in the flexible intracellular loop, whereas the Bayliss group used frog Panx1 tagged with GFP at the C-terminus. This difference in tagging and species may have contributed to variations in basal activity. Second, the lipid compositions used for reconstitution were significantly different. In our experiments, we used POPE, POPG, and sphingomyelin, while the Bayliss group employed a mixture of brain phosphatidylcholine, total brain lipid extract, cholesterol, and phosphatidylinositol 4,5-bisphosphate. Given that the function of many ion channels is heavily influenced by lipid composition, these differences could have contributed to the observed discrepancy. Regardless, our functional reconstitution experiments clearly demonstrate that LPC stimulates YO-PRO-1 uptake in a dose-dependent manner, which forms the foundation of our interpretation.

Although our activity-guided fractionation led to the discovery of lysophospholipids as activators of Panx1 and Panx2, it is conceivable that other classes of activators may exist for these channels. Particularly, whole-cell patch-clamp experiments would limit the target molecules to those acting externally, potentially overlooking molecules that act intracellularly. Additionally, our extraction and fractionation methods were biased towards relatively non-polar compounds, leaving open the possibility that hydrophilic molecules may also activate pannexins. Nevertheless, given the signaling roles of lysophospholipids, especially in immune responses where pannexin involvement has been well-documented, it makes sense that these signaling molecules trigger the activation of pannexin channels.

In conclusion, our study supports the idea that pannexins are key downstream players in lysophospholipid signaling. Our discovery connects purinergic signaling with lipid metabolism and bolsters the importance of the less-understood lysophospholipid-mediated signaling arm of the clinically impactful lipid mediator inflammatory pathway. Alongside the well-studied activation observed in dying cells, our study unveils another layer of activation mechanisms of pannexins in living cells.

## Materials and methods

**Key resources table**

| Reagent type (species) or resource | Designation | Source or reference | Identifiers | Additional information |
|---|---|---|---|---|
| Peptide, recombinant protein (*Homo sapiens*) | Pannexin1 | UniProt | Q96RD7 | Coding DNA was synthesized and subcloned into multiple expression vectors |
| Peptide, recombinant protein (*Homo sapiens*) | Pannexin2 | UniProt | Q96RD6 | Coding DNA was synthesized and subcloned into multiple expression vectors |
| Peptide, recombinant protein (*Homo sapiens*) | Pannexin3 | UniProt | Q96QZ0 | Coding DNA was synthesized and subcloned into multiple expression vectors |
| Peptide, recombinant protein (*Xenopus tropicalis*) | Pannexin1 | UniProt | B3DLA5 | Coding DNA was synthesized and subcloned into multiple expression vectors |
| Peptide, recombinant protein (*Homo sapiens*) | Connexin-43 | UniProt | P17302 | Coding DNA was synthesized and subcloned into multiple expression vectors |
| Peptide, recombinant protein (*Homo sapiens*) | LRRC8A | UniProt | Q8IWT6 | Coding DNA was synthesized and subcloned into multiple expression vectors |
| Peptide, recombinant protein (*Caenorhabditis elegans*) | Innexin-6 | UniProt | Q9U3N4 | Coding DNA was synthesized and subcloned into multiple expression vectors |
| Recombinant DNA reagent | pIE2 (plasmid) | This paper | | Kawate lab plasmid, mammalian expression vector with IRES-GFP |
| Recombinant DNA reagent | pCNG-FB7 (plasmid) | This paper | | Kawate lab plasmid, insect cell expression vector |
| Recombinant DNA reagent | pEZT-BM (plasmid) | Addgene | 74099 | |
| Cell line (*Spodoptera frugiperda*) | Sf9 | ThermoFisher | 12659017 | RRID:CVCL_0549 |
| Cell line (*Trichoplusia ni*) | High Five | ThermoFisher | B85502 | RRID:CVCL_C190 |
| Cell line (*Homo-sapiens*) | HEK293 | ATCC | CRL-1573 | RRID:CVCL_0045 |
| Cell line (*Homo-sapiens*) | HEK293T | ATCC | CRL-3216 | RRID:CVCL_0063 |
| Cell line (*Homo-sapiens*) | HEK293S GnTI- | ATCC | CRL-3022 | RRID:CVCL_A785 |
| Cell line (*Homo-sapiens*) | THP-1 | ATCC | TIB-202 | RRID:CVCL_0006 |
| Transfected construct (*Homo-sapiens*) | *PANX1*- shRNA #1 | BROAD Institute Genetic Perturbation Platform | TRCN0000155348 | Lentiviral construct to transfect and express the shRNA |
| Transfected construct (*Homo-sapiens*) | *PANX1*- shRNA #2 | BROAD Institute Genetic Perturbation Platform | TRCN0000154636 | Lentiviral construct to transfect and express the shRNA |
| Antibody | anti-FLAG M2 (Mouse monoclonal) | Millipore Sigma | F1804 | WB (1:2000) RRID:AB_262044 |
| Antibody | anti-actin (Mouse monoclonal) | Millipore Sigma | A4700 | WB (1:2000) RRID:AB_476730 |
| Antibody | anti-Panx1 (Rabbit monoclonal) | Cell Signaling Technologies | 91137 | WB (1:1000) RRID:AB_2800167 |
| Antibody | anti-cleaved IL-1β (Rabbit monoclonal) | Cell Signaling Technologies | 83186 | WB (1:1000) RRID:AB_2800010 |
| Antibody | anti-cleaved pro-IL-1β (Mouse monoclonal) | Cell Signaling Technologies | 12242 | WB (1:1000) RRID:AB_2715503 |
| Antibody | anti-NLRP3 (Rabbit monoclonal) | Cell Signaling Technologies | 15101 | WB (1:1000) RRID:AB_2722591 |
| Antibody | anti-Caspase 1 (Mouse monoclonal) | AdipoGen | AG-20B-0048 | WB (1:1000) RRID:AB_2490257 |
| Antibody | anti-rabbit IgG HRP-linked | Cell Signaling Technologies | 7074 | WB (1:5000) RRID:AB_2099233 |
| Antibody | anti-mouse IgG HRP-linked | Cell Signaling Technologies | 7076 | WB (1:5000) RRID:AB_330924 |

## Reagents

Detergents were purchased from Anatrace and lipids were purchased from Avanti Polar Lipids. Carbenoxolone (CBX), Arachidonic acid (AA), arachidonoylethanolamide (AEA), quercetin (QCT), chlorpromazine (CPZ), PP2, biotin and desthiobiotin and were purchased from Sigma-Aldrich. Prostaglandins (PGE2, PGI2) were purchased from Abcam. Porcine pancreas sPLA2 (P6534) was purchased from Millipore Sigma. Z-DEVD-FMK (ZVAD, 210344-95-9) was purchased from Santa Cruz Biotechnology. Mastoparan (MTP) was purchased from Santa Cruz Biotechnology (sc-200831) or Millipore Sigma (M5280). D-luciferin and luciferase were purchased from the Thermo Fisher Scientifc (A22066). Lysolipids were dissolved in 100% $CHCl_3$ (Lysophosphatidylcholine (LPC) and Lysosphingomyelin (LSM)) or 70% EtOH (Lysophosphatidylinonsitol (LPI), Lysophosphatidylethanolamine (LPE), and Lysophosphatidic acid (LPA)). Solvent was evaporated under a stream of $N_2$ and lipids stored at –20 °C. On the day of experiments, lipids were redissolved in EtOH (LPC, LSM) or 70% EtOH(LPI, LPE, LPA) and then diluted into the indicated buffer.

## Cell culture

HEK293 (CRL-1573) and HEK293S GnTI⁻ (CRL-3022) cell lines were purchased from the American Type Culture Collection (ATCC). The mycoplasma contamination test was confirmed to be negative at ATCC and therefore were not further authenticated. HEK293 cells were maintained in Dulbecco's modified Eagle medium (ThermoFisher Scientific) supplemented with 10% FBS (Corning Life Sciences) and 10 µg/ml gentamicin (Quality Biological) at 37 °C with 8% $CO_2$ in a humidified incubator. GnTI⁻ cells were maintained in FreeStyle 293 (Thermo Fisher Scientific) supplemented with 2.5% FBS at a shaking speed of 125 rpm at 37 °C with 8% $CO_2$ in a humidified incubator. Sf9 cells (Thermo Fisher Scientific) were maintained in Sf-900 III SFM (Thermo Fisher Scientific) and High Five cells (ThermoFisher Scientific) cells were maintained in ESF 921 (Expression Systems) at 27 °C and a shaking speed of 125 rpm. THP-1 cells (ATCC, TIB-202) were cultured in RPMI 1640 (Gibco, 11875–085) supplemented with 10% fetal bovine serum (FBS, Gibco, A52567-01, not heat-inactivated) without antibiotics and under a humidified atmosphere of 5% $CO_2$ at 37 °C. Cells were passed every 3–4 days, keeping their concentration to less than $1x10^6$ cells/mL. HEK293T cells were cultured in DMEM (Gibco, 11965–092) supplemented with 10% heat-inactivated calf serum (Sigma-Aldrich, C8056) without antibiotics and under a humidified atmosphere of 5% $CO_2$ at 37 °C. For *PANX1* shRNA stable knockdown, lentivirus was generated by co-transfection of pLKO.1 with shRNA sequences specific to human *PANX1* (TRCN0000155348 or TRCN0000154636, Millipore), psPAX2, and pMD2.G plasmids into HEK293T cells. The cell medium was collected and filtered through a 0.45 µm filter 48 hr after transfection and used to infect low-passage (less than 20) THP-1 cells. After 48 hr, infected cells were treated with puromycin (1 µg/mL, Goldbio) to select for stably incorporated shRNA constructs; cells were kept under selection for 3 weeks and then passed twice without puromycin before being used in experiments. Knockdown efficiency was validated by immunoblotting with pannexin-1 primary antibody (Cell Signaling Technologies, 91137, 1:1000 in 5% BSA-TBST).

## Expression constructs

DNA constructs encoding the amino acid sequences of the human Panx1 (hPanx1: gene ID: 24145), frog Panx1 (frPanx1: 100170473), human Panx2 (hPanx2: 56666), human Panx3 (hPanx3: 116337), human connexin 43 (hConx43: 2697), *C. elegans* Inx6 (ceInx6: 178231), and human LRRC8a (hLRRC8a: 56262) were synthesized (GenScript) and subcloned using a standard molecular cloning techniques into pIE2 vector for transient expression in HEK cells, pCNG-FB7 vector for insect cell expression, or pEZT-BM (Addgene: 74099) for infecting GnTI⁻ cells. The resulting insertion of extra residues were removed by PCR to maintain the native amino acid sequences. For pannexin western blotting, C-terminal FLAG tag was introduced by PCR. For hPanx1 purification from GnTI⁻ cells, the flexible intracellular loop (residues 175–182) were replaced with a StrepII tag by PCR. For mVenus quenching assays, the pEZT-BM vector was modified such that the multiple cloning site is followed by the internal ribosome entry site (IRES) from encephalomyocarditis virus (EMCV) that drives the expression of mVenus (GenBank ID: AAZ65844.1; kind gift from Matt Paszek at Cornell University). To enhance the ability of iodide to quench mVenus, two point mutations (H148Q and I152L) were introduced. For cellular YO-PRO-1 uptake assays, mVenus was replaced with mCherry (GenBank ID: AY678264.1; kind gift from Hiro Furukawa at CSHL). All constructs were verified by sanger sequencing.

## Whole-cell patch-clamp recordings

HEK cells or GnTI⁻ cells (passage number <40) were plated at low density onto 12 mm glass cover-slips in wells of a six-well plate (Greiner). Cells were transfected after 24 hr with ~700 ng plasmid DNA using FuGENE6 (Promega) according to the manufacturer's instructions, or infected with 5% (V/V) bacmam P2 virus. Recordings were obtained 40–60 hr after infection/transfection. Borosilicate glass pipettes (Harvard Apparatus) were pulled and heat polished to a final resistance of 2–4 MΩ and backfilled with (in mM) 147 NaCl, 10 EGTA, and 10 HEPES (adjusted to pH 7.0 with NaOH). For ion selectivity experiments, pipette solutions were identical except NaCl was replaced with NMDG-Cl or NaGluc. Patches were obtained in an external buffer containing (in mM) 147 NaCl, 2 KCl, 2 CaCl$_2$, 1 MgCl$_2$, 13 glucose, and 10 HEPES (adjusted to pH 7.3 with NaOH). For ion selectivity experiments, we exchanged external solutions before LPC-16:0 stimulation to a buffer containing NMDG-Cl (147 NMDG-Cl, 2 KCl, 2 CaCl$_2$, 1 MgCl$_2$, 13 glucose, and 10 HEPES adjusted to pH 7.3 with NaOH) or NaGluc (147 NaGluc, 2 KCl, 2 CaCl$_2$, 1 MgCl$_2$, 13 glucose, and 10 HEPES adjusted to pH 7.3 with NaOH). A rapid solution exchange system (RSC-200; Bio-Logic) was used for recordings in which patches were perfused with drugs or mouse fractions. Currents were recorded using an Axopatch 200B patch-clamp amplifier (Axon Instruments), filtered at 2 kHz (Frequency Devices), digitized with a Digidata 1440 A (Axon Instruments) with a sampling frequency of 10 kHz, and analyzed using the pCLAMP 10.5 software (Axon Instruments).

## Metabolomic screening

Frozen mouse liver tissues (C57BL6J mice; 3–6 months in age; 23 females and 24 males total used in this study) were extracted in 80% MeOH using a laboratory blender on dry ice. The resulting suspension was sonicated with a microtip probe sonicator (Qsonica Ultrasonic Processor, Model Q700) for 2 min (2 s on/off pulse cycle) on ice at 100% power. After removing the insoluble debris by centrifugation at 5250 RCF at 4 °C for 20 min, the supernatant was concentrated *in vacuo* in an SCP250EXP Speedvac Concentrator coupled to an RVT5105 Refrigerated Vapor Trap (Thermo Fisher Scientific), loaded onto Celite, and fractionated by medium pressure reverse-phase chromatography (30-gram C18 Combiflash RediSep Gold, Teledyne Isco). For the first round of fractions, an aqueous/acetonitrile solvent gradient was used at a flow rate of 20 mL/min, starting at 5% acetonitrile for 5 min and increasing to 100% acetonitrile over a period of 1 hr. For the second round of fractionation, the active fractions from the first round were combined and further fractionated by preparative reverse-phase chromatography using a Thermo Hypersil GOLD C18 column (10×250 mm/ 5 µm particle diameter; 25005–259070) with a 0.1% aqueous formic acid/acetonitrile gradient at a flow rate of 5 mL/min. After fractions were collected and the solvent was removed *in vacuo*, the dried fractions were stored at –20 °C. Each fraction was reconstituted in 5–25% DMSO and diluted 20-fold (first fractionation) or 300-fold (second fractionation) with the external buffer for the whole-cell patch-clamp experiments. Liquid chromatography was performed on a Thermo Vanquish Horizon HPLC system controlled by Chromeleon software (Thermo Fisher Scientific) coupled to an Orbitrap Q-Exactive HF mass spectrometer controlled by Xcalibur software (Thermo Fisher Scientific) outfitted with a heated electrospray ionization (HESI-II) probe. HPLC separation was achieved using a Thermo Hypersil GOLD C18 column (2.1×150 mm 1.9 µm particle size; 25002–152130) maintained at 40 °C. Solvent A: 0.1% formic acid in water; Solvent B: 0.1% formic acid in acetonitrile. A/B gradient started at 1% B for 3 min after injection and increased linearly to 98% B at 20 min, followed by 5 min at 98% B, then back to 1% B over 0.1 min and finally held at 1% B for the remaining 2.9 min (28 min total method time). Mass spectrometer parameters: spray voltage, −3.0 kV/+3.5 kV; capillary temperature 380 °C; probe heater temperature 400 °C; sheath, auxiliary, and sweep gas, 60, 20, and 2 AU, respectively; S-Lens RF level, 50; resolution, 120,000 at *m/z* 200; AGC target, 3E6. HPLC-HRMS data were analyzed using Metaboseek software with default settings after file conversion to the mzXML format via MSConvert (version 3.0, ProteoWizard)(*Helf et al., 2022*). The criteria used to define a feature of interest: minimum 10-fold enrichment in the active fractions (#7 and #8) relative to neighboring fractions (#6 and #9) and a mean intensity of 5,000,000 arbitrary units (AU) in ES +or 2,000,000 AU in ES- for a given feature of interest in the active fractions. The resulting feature lists were manually curated to remove isotopes, adducts, and fragments, yielding 12 metabolites of interest (*Table 1*).

**Table 1.** Enriched compounds in fractions 7 and 8.

| RT (min) | [M]+ | Obs. _m/z_ | Calc. _m/z_ | Error (ppm) | [M]- | Obs. _m/z_ | Calc. _m/z_ | Error (ppm) | Formula | ID |
|---|---|---|---|---|---|---|---|---|---|---|
| 15.05 | ND | - | - | - | M-H | 229.1808 | 229.1809 | 0.46 | C13H26O3 | Unknown. Likely hydroxylated fatty acid |
| 15.1 | M+H | 454.2917 | 454.2928 | 2.37 | M-H | 452.2782 | 452.2783 | 0.24 | C21H44NO7P | LPE(0/16:0)* |
| 15.12 | ND | - | - | - | M-H | 243.1964 | 243.1966 | 0.54 | C14H28O3 | Unknown. Likely hydroxylated fatty acid |
| 15.16 | M+H | 335.3047 | 335.3057 | 3.09 | ND | - | - | - | C21H38N2O | Unknown |
| 15.25 | ND | - | - | - | M-H | 241.1809 | 241.1809 | 0.06 | C14H26O3 | Unknown. Likely hydroxylated fatty acid |
| 15.32 | ND | - | - | - | M-H | 243.1964 | 243.1966 | 0.51 | C14H28O3 | Unknown. Likely hydroxylated fatty acid |
| 15.38 | M+H | 454.2918 | 454.2928 | 2.27 | M-H | 452.2781 | 452.2783 | 0.30 | C21H44NO7P | LPE(16:0/0) |
| 15.39 | M+H | 496.3388 | 496.3398 | 1.85 | M+formate | 540.3296 | 540.3307 | 2.03 | C24H51NO7P | LPC(0/16:0) |
| 15.56 | M+H | 546.3548 | 546.3554 | 1.15 | M+formate | 590.3465 | 590.3463 | –0.21 | C28H53NO7P | LPC(20:3/0)* |
| 15.68 | M+H | 496.3389 | 496.3398 | 1.80 | M+formate | 540.3297 | 540.3307 | 1.89 | C24H51NO7P | LPC(16:0/0) |
| 15.76 | M+H | 522.3548 | 522.3554 | 1.25 | M+formate | 566.3469 | 566.3463 | –0.96 | C26H53NO7P | LPC(0/18:1)* |
| 16.01 | M+H | 522.3546 | 522.3554 | 1.65 | M+formate | 566.3468 | 566.3463 | –0.86 | C26H53NO7P | LPC(18:1/0) |
| 21.86 | M+H (?) | 536.1646 | - | - | ND | - | - | - | ? | Unknown† |

ND; not detected.

*Predicted metabolite based on molecular formula and MS/MS fragmentation.

†Likely polysiloxane or similar contaminant. Detected most strongly in fraction 8, but additionally in fractions 4 and 5.

## Secretomics for LPC-16:0-stimulated Panx1

HEK293 GnTI⁻ cells were cultured on poly-D-lysine coated 6-well plates (Corning) and infected with bacmam P2 virus to induce expression of either mCherry vector alone or mCherry +Panx1. Twelve hours after the infection, 5 mM sodium butyrate was added to boost Panx1 expression and incubated for another 24 hr. Cells were washed with warmed assay buffer (147 NaCl, 10 HEPES, 13 Glucose, 2 KCl, 2 CaCl₂, 1 MgCl₂, pH 7.3 (mM)), then allowed to equilibrate in fresh assay buffer for five minutes. Cells were incubated with LPC-16:0 (final 10 μM) or an equivalent volume of assay buffer (vehicle control) for 45 min and the conditioned media was harvested by centrifugation at 1000 x _g_ for 5 min at 4 °C. Conditioned media samples were lyophilized for 24–30 hr using a VirTis BenchTop 4 K Freeze Dryer and then resuspended in 2 mL MeOH by vortexing and water bath sonication. The methanolic extracts were centrifuged at 4000 x _g_, for 5 min at 4 °C, and the resulting clarified supernatants were transferred to clean 4 mL glass vials, which were concentrated to dryness in an SC250EXP SpeedVac Concentrator coupled to an RVT5105 Refrigerated Vapor Trap (Thermo Fisher Scientific). Samples were resuspended in 0.6 mL MeOH and again concentrated to dryness as described above. Samples were finally resuspended in 70 μL MeOH, centrifuged at 4000 x _g_ for 10 min at 4 °C and the clarified supernatant used for MS analysis.

Normal-phase chromatography was performed using the same system as described above. Methanolic extracts were separated on a Waters XBridge Amide column (150 mm × 2.1 mm, particle size 3.5 μm; Catalog no. 186004861) maintained at 40 °C with a flow rate of 0.5 mL/min. Solvent A: 0.1% v/v ammonium formate in 90% acetonitrile/10% water; solvent B: 0.1% v/v ammonium formate in 30% acetonitrile/70% water. The LC method started at 1% B from 0 to 3 minutes, then increased linearly from 1% B to 60% B from 3 to 20 min, then increased linearly to 100% B from 20 to 26 min,

followed by 5 min isocratic at 100% B from 26 to 28 min, then returned to 1% B isocratic from 28 to 31 min to re-equilibrate prior to the next injection.

Mass spectrometer parameters: spray voltage, –3.0 kV/+3.5 kV; capillary temperature, 380 °C; sheath gas, 60 AU; auxiliary gas, 20 AU; sweep gas, 1 AU; probe heater temperature, 400 °C; S-lens RF level 50. Full MS-SIM: resolution, 120,000 or 140,000 at $m/z$ 200; AGC target, $5 \times 10^6$; scan range, 70–1000 m/z. Full MS/dd-MS2: MS1 resolution, 60,000 at $m/z$ 200; AGC target, $3 \times 10^6$; Scan range, 117–1000 m/z MS2 resolution, 30,000 at m/z 200; AGC target 5e5; maximum injection time 100ms; isolation window, 1.0 m/z; stepped normalized collision energy (NCE) 10, 30; dynamic exclusion, 3.0 s; Loop count, 10.

HPLC-HRMS data were analyzed using Metaboseek software as described above. For the volcano plot in *Figure 3B*, blank subtraction was performed by removing any feature less than five-fold more abundant in Panx1-expressing cells treated with LPC-16:0 relative to process blank injections. Features were further culled by removing any feature that did not have an accurate (<5 ppm) $m/z$ match to any known compound in the Human Metabolome Database (*Wishart et al., 2022*) using the *mzMatch* feature table analysis in Metaboseek. The resulting feature list of 517 features was grouped according to treatment and analyzed by unpaired, two-sided *t*-test with no multiple testing correction. Four independent experiments were performed. Compounds reported as differential were confirmed with authentic standards using their molecular ions detected in negative electrospray ionization for quantification. In the case of methylthioadenosine, we noted significant in-source fragmentation to yield a fragment corresponding to the mass of adenine.

## Cell surface biotinylation

Cell surface biotinylation, pulldown and subsequent immunoblotting were performed as described previously (*Michalski et al., 2018*). Briefly, HEK293 or HEK293 GnTI⁻ cells were plated onto a six-well plate and transfected using JetPrime (Polyplus) with 2.5 µg of FLAG-tagged pIE2-pannexin constructs. Two days post-transfection, cells were harvested washed in PBS. Surface membrane proteins were biotin-labeled by resuspending cells with 0.5 mg/ml sulfo-NHS-SS-biotin (Thermo Fisher Scientific) for 40 min at 4 °C. The reaction was quenched by washing cells twice with PBS supplemented with 50 mM $NH_4Cl$, followed by a final wash with 1 mL PBS. Cells were lysed in RIPA buffer (150 mM NaCl, 3 mM MgCl2, 1% NP-40, 0.5% deoxycholate (Anatrace), 0.1% SDS, 20 mM HEPES pH to 7.4 with NaOH) supplemented with 1 x protease inhibitor cocktail (Thermo Fisher Scientific) and rotated for 30 min. The lysate was clarified by centrifugation at 21,000 x $g$ for 15 min and the supernatant was recovered. Streptactin sepharose high-performance resin (GE Healthcare) was added to the lysates and rotated for 2 hr and 30 min. Samples were washed six times and the biotinylated proteins were eluted by incubating resin with 1.5 x SDS sample buffer supplemented with 75 mM DTT for 30 min at 55 °C with intermittent vortexing. Anti-FLAG (1:2000; clone M2), or anti-actin monoclonal antibodies (1:2000; line AC-40), were used to detect the target proteins by western blot.

## Cellular pannexin activity assay

HEK293 or HEK293 GnTI⁻ cells were plated onto 96-well poly-D-lysine coated, black-walled plates (Corning) at ~90% confluency and infected with 10% (v/v) BacMam P2 virus made with the pEZT-BM constructs encoding pannexins (and mVenus for the quench assays). Sodium butyrate (5 mM) was added to the media 10–12 hr after infection and the pannexin activity assays were performed 40–48 hr after infection (36–40 hr for hPanx1 expressed GnTI⁻ cells). Growth media was replaced with Assay buffer containing in mM: 147 NaI (NaCl for YO-PRO-1 uptake assays), 2 $CaCl_2$, 2 KCl, 1 $MgCl_2$, 10 HEPES, 13 Glucose, pH 7.3. For the stimulation with LPA, Assay buffer was modified to include: 147 NaI, 2 KCl, 1 EGTA, 10 HEPES, 13 Glucose, pH 7.3 for preventing precipitation. After 10 min equilibration (for mVenus-quench) or 5 min equilibration with 5 µM YO-PRO-1 (for YO-PRO-1 uptake), the cells were stimulated with various compounds and enzymes as indicated in the figures. Inhibitors were added concomitantly or 5 min after the stimulation for sPLA2. Pannexin activity was monitored by measuring fluorescence using a plate reader (Biotek Synergy2) at 480 nm (excitation) and 528 nm (emission) wavelengths with 20 nm bandwidth. The maximum quench/fluorescence was obtained with 1% Triton-X100 or Tween-20. The percent of quenching $Q(t)$ was calculated according to the following formula:

$$Q\left(t\right) = 100 * \frac{F\left(t\right)F_i}{F_f F_i}$$

where $F_i$ is the initial fluorescence at 10 min, $F_f$ is the final fluorescence following addition of the detergent, and $F(t)$ is the fluorescence value at each time point. Each condition was tested in experimental triplicate and the resulting averaged $Q(t)$ was used to obtain mVenus quenching or YO-PRPO-1 uptake rates in the initial linear range using the SLOPE function in Microsoft Excel. The dose response curves were obtained by fitting the plots with the Hill equation in Mathematica after subtracting the background fluorescence observed from the cells transfected with the vector alone.

## ATP release assay

HEK293 or HEK293 GnTI⁻ cells were plated onto poly-D-lysine coated, 96-well white-walled plates (Corning) at ~90% confluency and infected with 10% (v/v) BacMam P2 virus made with the pEZT-BM constructs encoding pannexins. Cells were washed with the external patch-clamp buffer and incubated for 10 min with D-luciferin (0.5 µM) and luciferase (1.3 µg/mL) using ATP Determination Kit (ThermoFisher Scientific). Three minutes after the addition of LPC-16:0 (30 µM) in the presence or absence of CBX (50 µM), ATP released in the external buffer was measured by following the luminescence using a plate reader (Biotek Synergy2). Luminescence was normalized to the maximum value obtained with 1% Triton-X100.

## Panx1 purification

Panx1 was purified as described previously (*Michalski et al., 2020*). Briefly, Panx1 expressing cells (GnTI- for hPanx1 and HighFive cells for frPanx1-ΔLC+GS) were harvested by centrifugation and broken by nitrogen cavitation (4635 cell disruption vessel; Parr Instruments). The membrane fraction was recovered by centrifugation at 12,000 x *g* for 10 min followed by at 185,000 × *g* for 45 min in PBS supplemented with a protease inhibitor cocktail (2.0 µg/mL leupeptin, 8.0 µg/mL aprotinin, 2.0 µg/mL pepstatin, and 0.5 mM phenylmethylsulfonyl fluoride). Membranes were solubilized in S buffer (PBS, protease inhibitor cocktail, 10% glycerol, and 1% C12E8) for 60 minutes. Following ultracentrifugation at 185,000 x *g* for 45 min, the supernatant was incubated with Strep-Tactin Sepharose resin or Strep-Tactin XT resin (Cytiva) for 60 min. The resin was washed with 10 column volumes of Wash buffer (150 mM NaCl, 100 mM Tris-HCl pH 8.0, 10% glycerol, and 0.27% C12E8) and the bound protein was eluted with five column volumes of Wash buffer containing 2.5 mM desthiobiotin (for Sepharose) or 25 mM biotin (for XT resin). Eluted Panx1 was further purified by size-exclusion chromatography (Superdex 200; Cytiva) in SEC buffer (150 mM NaCl, 20 mM Tris-HCl pH 8.0, and 0.27% C12E8). A single monodisperse Panx1 peak was collected for functional reconstitution into liposomes or into lipid nanodiscs for cryo-EM studies. All steps were carried out at 4 °C or on ice.

## Functional reconstitution of Panx1

Empty liposomes composed of 1-palmitoyl-2-oleoyl-sn-glycero-3-phosphoethanolamine (POPE), 1-P almitoyl-2-oleoyl-sn-glycero-3-phosphoglycerol (POPG), and sphingomyelin (SM; brain extract) were resuspended in Reconstitution buffer (in mM; 50 Tris-HCl, 150 NaCl, 0.1 EGTA, pH 7.4) to a concentration of 10 mg/mL and extruded 13 times through a 400 nm polycarbonate filter before use. Following incubation with 0.65% (w/v) DDM for 15 min, purified Panx1 was added at 50:1 (hPanx1) or at 100:1 (frPanx1-ΔLC+GS) ratio and incubated for 30 min. Detergents were removed by Bio-beads SM2 (Bio-Rad) and 20 bp oligo DNAs (40 µM each) were incorporated by freeze-thaw and extrusion through a polycarbonate filter. Unincorporated DNA was digested by incubating the liposomes with 0.2 mg/mL DNAse I in the presence of 5 mM MgCl₂ for 60 min, followed by centrifugation at 280,000 × *g* for 20 min and resuspension in Reconstitution buffer. DNA-incorporated proteoliposomes (50 µg/well) were dispensed into 96-well Poly-D-lysine coated black-walled plates (Corning). Upon LPC-16:0 and YO-PRO-1 (60 nM) application, Panx1 activity was monitored by measuring fluorescence using a plate reader (Biotek Synergy2) at 480 nm (excitation) and 528 nm (emission) wavelengths with 20 nm bandwidth. The maximum fluorescence was obtained with 1% Triton-X100. Quantification of Panx1 activity was performed as described above.

## Canine synovial fluid sample collection

Synovial fluid was collected from 22 different client-owned dogs presenting to the Cornell University Hospital for Animals. The Cornell University Veterinary Clinical Studies Committee approved the project (#100121–16), and all the dog owners gave informed consent to use samples from their dogs. Dogs were aged 6.5 (0.67–10) years (median [range]) and consisted of neutered females (n=13), neutered males (n=8), and an intact male (n=1). Represented breeds included mixed breed (n=8), Labrador retriever (n=5), Golden retriever (n=2), and n=1 each of Akita, Australian shepherd, Bernese mountain dog, Chesapeake Bay retriever, German shepherd dog, German shorthaired pointer, and pit bull terrier. All dogs had a history of joint pain of at least 1 month duration secondary to developmental, degenerative, and/or inflammatory disease (e.g. degeneration and rupture of the cranial cruciate ligament, osteochondrosis dissecans, osteoarthritis, septic arthritis). Synoviocentesis was performed under sedation or general anesthesia for either diagnostic or therapeutic purposes (e.g. cytology, bacterial culture, or injection of pharmaceuticals [after sample collection], n=8), or before the start of orthopedic surgery on the joint (n=14); an aliquot was reserved for this study. Surgeries included tibial plateau-leveling osteotomy (TPLO) with (n=6) or without (n=3) meniscectomy, arthroscopy (n=2), multiple procedures on one stifle joint (n=2), or implant removal with meniscectomy (n=1). Samples were collected from the stifle (i.e. femorotibial or knee joint, n=17), elbow (n=8), and shoulder (n=4); 1 sample was collected from either the elbow or shoulder but not labeled. Using standard aseptic technique, at least 500 µL were collected into a 2.0 mL polypropylene microcentrifuge tube and frozen at –80 °C within 30 min. There is currently no single validated scale for grading severity of pain of all etiologies in dogs. Therefore, before anesthetic drugs were administered, and before the synovial fluid was collected, a single investigator (a veterinary anesthesiologist experienced in assessing pain in dogs, J.M.B.) used a simple descriptive scale to grade the severity of pain in the joint to be sampled; pain was assessed as mild, moderate, or severe. Clinical signs of pain included lameness, behavioral reaction upon palpation or manipulation of the joint (i.e. struggling, vocalizing), and impaired ability to perform normal activities, such as running or using stairs.

## IL-1β release assay using THP-1 cells

IL-1β response of PMA-differentiated THP-1 cells to treatment with LPC-16:0. THP-1 cells were seeded uniformly at a density of $1 \times 10^6$ cells per well in a 24-well plate with 1 mL 10% FBS-RPMI supplemented with phorbol 12-myristate 13-acetate (PMA, 100 ng/mL). Empty pLKO.1 vector lentivirus-transduced THP-1 cells were used as the control. After 48 hr of differentiation, the media was changed to 1 mL of 10% FBS-RPMI without PMA. After 24 hr of rest, the media was replaced with 0.5 mL of 10% FBS-RPMI supplemented with or without lipopolysaccharide from *Escherichia coli* O111:B4 (LPS, 200 ng/mL, Sigma, L2630). After 3.5 hr of priming in the incubator, each well was gently washed thrice with 0.5 mL of 37 °C serum-free RPMI (SF-RPMI). Throughout the washing procedure, a 20 mM solution of LPC-16:0 (Sigma, L5254) in PBS in a polypropylene tube was incubated at 37 °C in a water bath. The 20 mM LPC-16:0 solution was removed from the water bath and vigorously vortexed, and then an aliquot was diluted in 37 °C SF-RPMI to a final concentration of 50 µM with gentle end-over-end mixing. The PBS wash buffer was aspirated from each well and gently replaced with 380 µL of the LPC-16:0-containing media; prior to each addition, the LPC-16:0-containing media was pipetted up-and-down three times to minimize loss of LPC-16:0 due to its potential adsorption to the polypropylene pipette tips. The plate was placed in the incubator for 1.5 hr. The supernatant media was collected and cleared of cells by centrifugation at $5000 \times g$ at 4 °C for 5 min. A 100 µL aliquot of cleared supernatant was collected for each treatment group; 20 µL of 6 x Laemmli buffer was added to each aliquot, which were then heated at 95 °C for 5 min. Immediately after removing the supernatant media from the wells of the plate, each well was washed once with 0.5 mL of ice-cold PBS. Cells were lysed by the addition of lysis buffer (25 mM Tris pH 8.0, 150 mM NaCl, 10% glycerol, 1% NP-40, protease inhibitor cocktail) to each well followed by shaking at 4 °C for 45 min. Total protein of the cleared (17,000 × *g*, 4 °C, 20 min) cell lysates was quantified by the Bradford assay (Pierce Bradford Protein Assay Kit, Thermo, 23200). A 100 µL aliquot of cleared cell lysate was collected for each treatment group; 20 µL of 6 x Laemmli buffer was added to each aliquot, which were then heated at 95 °C for 5 min. Media supernatants and cell lysates were analyzed by immunoblotting (12% SDS-PAGE, 200 V, 50 min; block 1 hr at room temperature in 5% BSA-TBST; transfer to 0.2 µM PVDF), with gel loading volume normalized to the total protein content of cell lysates to account for any potential loss of cells following the

three cycles of PBS washing. Blots were visualized on a BioRad ChemiDoc MP instrument using chemiluminescence (Clarity Max Western ECL Substrate [BioRad, 1705062 S] or Pierce ECL Western Blotting Substrate [Thermo, 32106]). The following primary antibodies were used at 1:1000 dilution at 4 °C overnight (5% BSA-TBST): Panx1 (91137, Cell Signaling Technologies [CST]), cleaved IL-1β (83186, CST), pro-IL-1β (12242, CST), NLRP3 (15101, CST), and caspase 1 (AG-20B-0048-C100, AdipoGen). The following secondary antibodies were used at room temperature for 1 hr at 1:5000 dilution (5% BSA-TBST): anti-rabbit IgG HRP-linked (7074 V, CST) and anti-mouse IgG HRP-linked (7076, CST).

## Materials availability

All materials, including plasmids generated in this study, are available from the authors upon request.

## Acknowledgements

We thank O Boudker, K Swartz, J Davis, A Alouani, Biophysics Colab, and the members the Kawate labs for helpful discussions and comments for the manuscript; W Greentree and M Linder for providing mouse tissues; S Webb for the help on joint fluid sample collection.

## Additional information

### Competing interests

The authors declare that no competing interests exist.

### Funding

| Funder | Grant reference number | Author |
|---|---|---|
| National Institutes of Health | R01GM114379 | Toshimitsu Kawate |
| National Institutes of Health | R35GM131877 | Frank C Schroeder |
| National Institutes of Health | T32GM008267 | Kevin Michalski |
| National Science Foundation | DBI1659534 | Lydia Kramer |
| Cornell Margaret and Richard Riney Canine Health Center Research Grants Program | | Erik Henze Jordyn M Boesch Toshimitsu Kawate |
| National Institutes of Health | R01DK107451 | Hening Lin |

The funders had no role in study design, data collection and interpretation, or the decision to submit the work for publication.

### Author contributions

Erik Henze, Data curation, Formal analysis, Validation, Investigation, Methodology, Writing – original draft; Russell N Burkhardt, Tyler J Schwertfeger, Formal analysis, Validation, Investigation, Methodology, Writing – review and editing; Bennett William Fox, Data curation, Formal analysis, Validation, Investigation, Visualization, Methodology, Writing – review and editing; Eric Gelsleichter, Formal analysis, Validation, Investigation, Visualization, Methodology, Writing – review and editing; Kevin Michalski, Investigation, Writing – review and editing; Lydia Kramer, Resources, Investigation, Writing – review and editing; Margret Lenfest, Resources, Investigation, Methodology, Writing – review and editing; Jordyn M Boesch, Resources, Formal analysis, Supervision, Funding acquisition, Validation, Writing – review and editing; Hening Lin, Supervision, Funding acquisition, Writing – review and editing; Frank C Schroeder, Conceptualization, Resources, Supervision, Funding acquisition, Validation, Writing – review and editing; Toshimitsu Kawate, Conceptualization, Resources, Data curation,

Formal analysis, Supervision, Funding acquisition, Validation, Investigation, Visualization, Methodology, Writing – original draft, Project administration, Writing – review and editing

**Author ORCIDs**
Jordyn M Boesch http://orcid.org/0000-0002-2032-1202
Hening Lin http://orcid.org/0000-0002-0255-2701
Toshimitsu Kawate https://orcid.org/0000-0002-5005-2031

Joint Public Review: https://doi.org/10.7554/eLife.107067.2.sa1
Author response https://doi.org/10.7554/eLife.107067.2.sa2

## Additional files

**Supplementary files**
MDAR checklist

**Data availability**
All data are available in the figures or in Source data.

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
