## [Editor Report · eLife Assessment]

Pannexin (Panx) channels are a family of poorly understood large-pore channels that mediate the release of substrates like ATP from cells, yet the physiological stimuli that activate these channels remain poorly understood. The study by Henze et al. describes an elegant approach wherein activity-guided fractionation of mouse liver led to the discovery that lysophospholipids (LPCs) activate Panx1 and Panx2 channels expressed in cells or reconstituted into liposomes. The authors provide **compelling** evidence that LPC-mediated activation of Panx1 is involved in joint pain and that Panx1 channels are required for the established effects of LPC on inflammasome activation in monocytes, suggesting that Panx channels play a role in inflammatory pathways. Overall, this **important** study reports a previously unanticipated mechanism wherein LPCs directly activate Panx channels. The work will be of interest to scientists investigating phospholipids, Panx channels, purinergic signalling and inflammation.

[Editors' note: this paper was reviewed and curated by Biophysics Colab]

---

## [Referee Report · Joint Public Review]

Pannexin (Panx) hemichannels are a family of heptameric membrane proteins that form pores in the plasma membrane through which ions and relatively large organic molecules can permeate. ATP release through Panx channels during the process of apoptosis is one established biological role of these proteins in the immune system, but they are widely expressed in many cells throughout the body, including the nervous system, and likely play many interesting and important roles that are yet to be defined. Although several structures have now been solved of different Panx subtypes from different species, their biophysical mechanisms remain poorly understood, including what physiological signals control their activation. Electrophysiological measurements of ionic currents flowing in response to Panx channel activation have shown that some subtypes can be activated by strong membrane depolarization or caspase cleavage of the C-terminus. Here, Henze and colleagues set out to identify endogenous activators of Panx channels, focusing on the Panx1 and Panx2 subtypes, by fractionating mouse liver extracts and screening for activation of Panx channels expressed in mammalian cells using whole-cell patch clamp recordings. The authors present a comprehensive examination with robust methodologies and supporting data that demonstrate that lysophospholipids (LPCs) directly Panx-1 and 2 channels. These methodologies include channel mutagenesis, electrophysiology, ATP release and fluorescence assays, and molecular modelling. Mouse liver extracts were initially used to identify LPC activators, but the authors go on to individually evaluate many different types of LPCs to determine those that are more specific for Panx channel activation. Importantly, the enzymes that endogenously regulate the production of these LPCs were also assessed along with other by-products that were shown not to promote pannexin channel activation. In addition, the authors used synovial fluid from canine patients, which is enriched in LPCs, to highlight the importance of the findings in pathology. Overall, we think this is likely to be an important study because it provides strong evidence that LPCs can function as activators of Panx1 and Panx2 channels, linking two established mediators of inflammatory responses and opening an entirely new area for exploring the biological roles of Panx channels. This study provides an excellent foundation for future studies and importantly provides clinical relevance.

[Editors' note: this paper has been through two rounds of review and revisions, available here: https://sciety.org/articles/activity/10.1101/2023.10.23.563601]

---

## [Author Response]

(This author response relates to the first round of peer review by Biophysics Colab. Reviews and responses to both rounds of review are available here: https://sciety.org/articles/activity/10.1101/2023.10.23.563601.)

**General Assessment:**
Pannexin (Panx) hemichannels are a family of heptameric membrane proteins that form pores in the plasma membrane through which ions and relatively large organic molecules can permeate. ATP release through Panx channels during the process of apoptosis is one established biological role of these proteins in the immune system, but they are widely expressed in many cells throughout the body, including the nervous system, and likely play many interesting and important roles that are yet to be defined. Although several structures have now been solved of different Panx subtypes from different species, their biophysical mechanisms remain poorly understood, including what physiological signals control their activation. Electrophysiological measurements of ionic currents flowing in response to Panx channel activation have shown that some subtypes can be activated by strong membrane depolarization or caspase cleavage of the C-terminus. Here, Henze and colleagues set out to identify endogenous activators of Panx channels, focusing on the Panx1 and Panx2 subtypes, by fractionating mouse liver extracts and screening for activation of Panx channels expressed in mammalian cells using whole-cell patch clamp recordings. The authors present a comprehensive examination with robust methodologies and supporting data that demonstrate that lysophospholipids (LPCs) directly Panx-1 and 2 channels. These methodologies include channel mutagenesis, electrophysiology, ATP release and fluorescence assays, molecular modelling, and cryogenic electron microscopy (cryo-EM). Mouse liver extracts were initially used to identify LPC activators, but the authors go on to individually evaluate many different types of LPCs to determine those that are more specific for Panx channel activation. Importantly, the enzymes that endogenously regulate the production of these LPCs were also assessed along with other by-products that were shown not to promote pannexin channel activation. In addition, the authors used synovial fluid from canine patients, which is enriched in LPCs, to highlight the importance of the findings in pathology. Overall, we think this is likely to be a landmark study because it provides strong evidence that LPCs can function as activators of Panx1 and Panx2 channels, linking two established mediators of inflammatory responses and opening an entirely new area for exploring the biological roles of Panx channels. Although the mechanism of LPC activation of Panx channels remains unresolved, this study provides an excellent foundation for future studies and importantly provides clinical relevance.

We thank the reviewers for their time and effort in reviewing our manuscript. Based on their valuable comments and suggestions, we have made substantial revisions. The updated manuscript now includes two new experiments supporting that lysophospholipid-triggered channel activation promotes the release of signaling molecules critical for immune response and demonstrates that this novel class of agonist activates the inflammasome in human macrophages through endogenously expressed Panx1. To better highlight the significance of our findings, we have excluded the cryo-EM panel from this manuscript. We believe these changes address the main concerns raised by the reviewers and enhance the overall clarity and impact of our findings. Below, we provide a point-by-point response to each of the reviewers’ comments.

**Recommendations:**
(1) The authors present a tremendous amount of data using different approaches, cells and assays along with a written presentation that is quite abbreviated, which may make comprehension challenging for some readers. We would encourage the authors to expand the written presentation to more fully describe the experiments that were done and how the data were analysed so that the 2 key conclusions can be more fully appreciated by readers. A lot of data is also presented in supplemental figures that could be brought into the main figures and more thoroughly presented and discussed.

We appreciate and agree with the reviewers’ observation. Our initial manuscript may have been challenging to follow due to our use of both wild-type and GS-tagged versions of Panx1 from human and frog origins, combined with different fluorescence techniques across cell types. In this revision, we used only human wild-type Panx1 expressed in HEK293S GnTI- cells, except for activity-guided fractionation experiments, where we used GS-tagged Panx1 expressed in HEK293 cells (Fig. 1). For functional reconstitution studies, we employed YO-PRO-1 uptake assays, as optimizing the Venus-based assay was challenging. We have clarified these exceptions in the main text. We think these adjustments simplify the narrative and ensure an appropriate balance between main and supplemental figures.

(2) It would also be useful to present data on the ion selectivity of Panx channels activated by LPC. How does this compare to data obtained when the channel is activated by depolarization? If the two stimuli activate related open states then the ion selectivity may be quite similar, but perhaps not if the two stimuli activate different open states. The authors earlier work in eLife shows interesting shifts in reversal potentials (Vrev) when substituting external chloride with gluconate but not when substituting external sodium with N-methyl-D-glucamine, and these changed with mutations within the external pore of Panx channels. Related measurements comparing channels activated by LPC with membrane depolarization would be valuable for assessing whether similar or distinct open states are activated by LPC and voltage. It would be ideal to make Vrev measurements using a fixed step depolarization to open the channel and then various steps to more negative voltages to measure tail currents in pinpointing Vrev (a so called instantaneous IV).

We fully agree with the reviewer on the importance of ion selectivity experiments. However, comparing the properties of LPC-activated channels with those activated by membrane depolarization presented technical challenges, as LPC appears to stimulate Panx1 in synergy with voltage. Prolonged LPC exposure destabilizes patches, complicating G-V curve acquisition and kinetic analyses. While such experiments could provide mechanistic insights, we think they are beyond the scope of current study.

(3) Data is presented for expression of Panx channels in different cell types (HEK vs HEKS GnTI-) and different constructs (Panx1 vs Panx1-GS vs other engineered constructs). The authors have tried to be clear about what was done in each experiment, but it can be challenging for the reader to keep everything straight. The labelling in Fig 1E helps a lot, and we encourage the authors to use that approach systematically throughout. It would also help to clearly identify the cell type and channel construct whenever showing traces, like those in Fig 1D. Doing this systematically throughout all the figures would also make it clear where a control is missing. For example, if labelling for the type of cell was included in Fig 1D it would be immediately clear that a GnTI- vector alone control for WT Panx1 is missing as the vector control shown is for HEK cells and formally that is only a control for Panx2 and 3. Can the authors explain why PLC activates Panx1 overexpressed in HEK293 GnTl- cells but not in HEK293 cells? Is this purely a function of expression levels? If so, it would be good to provide that supporting information.

As mentioned above, we believe our revised version is more straightforward to digest. We have improved labeling and provided explanations where necessary to clarify the manuscript. While Panx1 expression levels are indeed higher in GnTI- than in HEK293 cells, we are uncertain whether the absence of detectable currents in HEK293 cells is solely due to expression levels. Some post-translational modifications that inhibit Panx1, such as lysine acetylation, may also impact activity. Future studies are needed to explore these mechanisms further.

(4) The mVenus quenching experiments are somewhat confusing in the way data are presented. In Fig 2B the y axis is labelled fluorescence (%) but when the channel is closed at time = 0 the value of fluorescence is 0 rather than 100 %, and as the channel opens when LPC is added the values grow towards 100 instead of towards 0 as iodide permeates and quenches. It would be helpful if these types of data could be presented more intuitively. Also, how was the initial rate calculated that is plotted in Fig 2C? It would be helpful to show how this is done in a figure panel somewhere. Why was the initial rate expressed as a percent maximum, what is the maximum and why are the values so low? Why is the effect of CBX so weak in these quenching experiments with Panx1 compared to other assays? This assay is used in a lot of experiments so anything that could be done to bolster confidence is what it reports on would be valuable to readers. Bringing in as many control experiments that have been done, including any that are already published, would be helpful.

We modified the Y-axis in Figure 2 to “Quench (%)” for clarity. The data reflects fluorescence reduction over time, starting from LPC addition, normalized to the maximal decrease observed after Triton-X100 addition (3 minutes), enabling consistent quenching value comparisons. Although the quenching value appears small, normalization against complete cell solubilization provides reproducible comparisons. We do not fully understand why CBX effects vary in Venus quenching experiments, but we speculate that its steroid-like pentacyclic structure may influence the lysophospholipid agonistic effects. As noted in prior studies (DOI: 10.1085/jgp.201511505; DOI: 10.7554/eLife.54670), CBX likely acts as an allosteric modulator rather than a simple pore blocker, potentially contributing to these variations.

(5) Could provide more information to help rationalize how Yo-Pro-1, which has a charge of +2, can permeate what are thought to be anion favouring Panx channels? We appreciate that the biophysical properties of Panx channel remain mysterious, but it would help to hear how a bit more about the authors thinking. It might also help to cite other papers that have measured Yo-Pro-1 uptake through Panx channels. Was the Strep-tagged construct of Panx1 expressed in GnTI- cells and shown to be functional using electrophysiology?

Our recent study suggest that the electrostatic landscape along the permeation pathway may influence its ion selectivity (DOI: 10.1101/2024.06.13.598903). However, we have not yet fully elucidated how Panx1 permeates both anions and cations. Based on our findings, ion selectivity may vary with activation stimulus intensity and duration. Cation permeation through Panx1 is often demonstrated with YO-PRO-1, which measures uptake over minutes, unlike electrophysiological measurements conducted over milliseconds to seconds. We referenced two representative studies employing YO-PRO-1 to assess Panx1 activity. Whole-cell current measurements from a similar construct with an intracellular loop insertion indicate that our STREP-tagged construct likely retains functional capacity.

(6) In Fig 5 panel C, data is presented as the ratio of LPC induced current at -60 mV to that measured at +110 mV in the absence of LPC. What is the rationale for analysing the data this way? It would be helpful to also plot the two values separately for all of the constructs presented so the reader can see whether any of the mutants disproportionately alter LPC induced current relative to depolarization activated current. Also, for all currents shown in the figures, the authors should include a dashed coloured line at zero current, both for the LPC activated currents and the voltage steps.

We used the ratio of LPC-induced current to the current measured at +110 mV to determine whether any of the mutants disproportionately affect LPC-induced current relative to depolarization-activated current. Since the mutants that did not respond to LPC also exhibited smaller voltage-stimulated currents than those that did respond, we reasoned that using this ratio would better capture the information the reviewer is suggesting to gauge. Showing the zero current level may be helpful if the goal was to compare basal currents, which in our experience vary significantly from patch to patch. However, since we are comparing LPC- and voltage-induced currents within the same patch, we believe that including basal current measurements would not add useful information to our study.

Given that new experiments included to further highlight the significance of the discovery of Panx1 agonists, we opted to separate structure-based mechanistic studies from this manuscript and removed this experiment along with the docking and cryo-EM studies.

(7) The fragmented NTD density shown in Fig S8 panel A may resemble either lipid density or the average density of both NTD and lipid. For example, Class7 and Class8 in Fig.S8 panel D displayed split densities, which may resemble a phosphate head group and two tails of lipid. A protomer mask may not be the ideal approach to separate different classes of NTD because as shown in Fig S8 panel D, most high-resolution features are located on TM1-4, suggesting that the classification was focused on TM1-4. A more suitable approach would involve using a smaller mask including NTD, TM1, and the neighbouring TM2 region to separate different NTD classes.

We agree with the reviewer and attempted 3D classification using multiple smaller masks including the suggested region. However, the maps remained poorly defined, and we were unable to confidently assign the NTD.

(8) The authors don’t discuss whether the LPC-bound structures display changes in the external part of the pore, which is the anion-selective filter and the narrower part of the pore. If there are no conformational changes there, then the present structures cannot explain permeability to large molecules like ATP. In this context, a plot for the pore dimension will be helpful to see differences along the pore between their different structures. It would also be clearer if the authors overlaid maps of protomers to illustrate differences at the NTD and the "selectivity filter."

Both maps show that the narrowest constriction, formed by W74, has a diameter of approximately 9 Å. Previous steered molecular dynamics simulations suggest that ATP can permeate through such a constriction, implying an ion selection mechanism distinct from a simple steric barrier.

(9) The time between the addition of LPC to the nanodisc-reconstituted protein and grid preparation is not mentioned. Dynamic diffusion of LPC could result in equal probabilities for the bound and unbound forms. This raises the possibility of finding the Primed state in the LPC-bound state as well. Additionally, can the authors rationalize how LPC might reach the pore region when the channel is in the closed state before the application of LPC?

We appreciate the reviewer’s insight. We incubated LPC and nanodisc-reconstituted protein for 30 minutes, speculating that LPC approaches the pore similarly to other lipids in prior structures. In separate studies, we are optimizing conditions to capture more defined conformations.

(10) In the cryo-EM map of the “resting” state (EMDB-21150), a part of the density was interpreted as NTD flipped to the intracellular side. This density, however, is poorly defined, and not connected to the S1 helix, raising concerns about whether this density corresponds to the NTD as seen in the “resting” state structure (PDB-ID: 6VD7). In addition, some residues in the C-terminus (after K333 in frog PANX1) are missing from the atomic model. Some of these residues are predicted by AlphaFold2 to form a short alpha helix and are shown to form a short alpha helix in some published PANX1 structures. Interestingly, in both the AF2 model and 6WBF, this short alpha helix is located approximately in the weak density that the authors suggest represents the “flipped” NTD. We encourage the authors to be cautious in interpreting this part as the “flipped” NTD without further validation or justification.

We agree that the density corresponding the extended NTD into the cytoplasm is relatively weak. In our recent study, we compared two Panx1 structures with or without the mentioned C-terminal helix and found evidence suggesting the likelihood of NTD extension (DOI: 10.1101/2024.06.13.598903). Nevertheless, to prevent potential confusion, we have removed the cryo-EM panel from this manuscript.

(11) Since the authors did not observe densities of bound PLC in the cryo-EM map, it is important to acknowledge in the text the inherent limitations of using docking and mutagenesis methods to locate where PLC binds.

Thank you for the suggestion. We have removed this section to avoid potential confusion.

Optional suggestions:(1) The authors used MeOH to extract mouse liver for reversed-phase chromatography. Was the study designed to focus on hydrophobic compounds that likely bind to the TMD? Panx1 has both ECD and ICD with substantial sizes that could interact with water soluble compounds? Also, the use of whole-cell recordings to screen fractions would not likely identify polar compounds that interact with the cytoplasmic part of the TMD? It would be useful for the authors to comment on these aspects of their screen and provide their rationale for fractionating liver rather than other tissues.

We have added a rationale in line 90, stating: “The soluble fractions were excluded from this study, as the most polar fraction induced strong channel activities in the absence of exogenously expressed pannexins.” Additionally, we have included a figure to support this rationale (Fig. S1A).

(2) The authors show that LPCs reversibly increase inward currents at a holding voltage of -60 mV (not always specified in legends) in cells expressing Panx1 and 2, and then show families of currents activated by depolarizing voltage steps in the absence of LPC without asking what happens when you depolarize the membrane after LPC activation? If LPCs can be applied for long enough without disrupting recordings, it would be valuable to obtain both I-V relations and G-V relations before and after LPC activation of Panx channels. Does LPC disproportionately increase current at some voltages compared to others? Is the outward rectification reduced by LPC? Does Vrev remain unchanged (see point above)? Its hard to predict what would be observed, but almost any outcome from these experiments would suggest additional experiments to explore the extent to which the open states activated by LPC and depolarization are similar or distinct.

Unfortunately, in our hands, the prolonged application of lysolipids at concentrations necessary to achieve significant currents tends to destabilize the patch. This makes it challenging to obtain G-V curves or perform the previously mentioned kinetic analyses. We believe this destabilization may be due to lysolipids’ surfactant-like qualities, which can disrupt the giga seal. Additionally, prolonged exposure seems to cause channel desensitization, which could be another confounding factor.

(3) From the results presented, the authors cannot rule out that mutagenesis-induced insensitivity of Panx channels to LPCs results from allosteric perturbations in the channels rather than direct binding/gating by LPCs. In Fig 5 panel A-C, the authors introduced double mutants on TM1 and TM2 to interfere with LPC binding, however, the double mutants may also disrupt the interaction network formed within NTD, TM1, and TM2. This disruption could potentially rearrange the conformation of NTD, favouring the resting closed state. Three double Asn mutants, which abolished LPC induced current, also exhibited lower currents through voltage activation in Fig 5S, raising the possibility the mutant channels fail to activate in response to LPC due to an increased energy barrier. One way to gain further insight would be to mutate residues in NTD that interact with those substituted by the three double Asn mutants and to measuring currents from both voltage activation and LPC activation. Such results might help to elucidate whether the three double Asn mutants interfere with LPC binding. It would also be important to show that the voltage-activated currents in Fig. S5 are sensitive to CBX?

Thank you for the comment, with which we agree. Our initial intention was to use the mutagenesis studies to experimentally support the docking study. Due to uncertainties associated with the presented cryo-EM maps, we have decided to remove this study from the current manuscript. We will consider the proposed experiments in a future study.

(4) Could the authors elaborate on how LPC opens Panx1 by altering the conformation of the NTDs in an uncoordinated manner, going from “primed” state to the “active” state. In the “primed” state, the NTDs seem to be ordered by forming interactions with the TMD, thus resulting in the largest (possible?) pore size around the NTDs. In contrast, in the “active” state, the authors suggest that the NTDs are fragmented as a result of uncoordinated rearrangement, which conceivably will lead to a reduction in pore size around NTDs (isn’t it?). It is therefore not intuitive to understand why a conformation with a smaller pore size represents an “active” state.

We believe the uncoordinated arrangement of NTDs is dynamic, allowing for potential variations in pore size during the activated conformation. Alternatively, NTD movement may be coupled with conformational changes in TM1 and the extracellular domain, which in turn could alter the electrostatic properties of the permeation pathway. We believe a functional study exploring this mechanism would be more appropriately presented as a separate study.

(5) Can the authors provide a positive control for these negative results presented in Fig S1B and C?

The positive results are presented in Fig. 1D and E.

(6) Raw images in Fig S6 and Fig S7 should contain units of measurement.

Thank you for pointing this out.

(7) It may be beneficial to show the superposition between primed state and activated state in both protomer and overall structure. In addition, superposition between primed state and PDB 7F8J.

We attempted to superimpose the cryo-EM maps; however, visually highlighting the differences in figure format proved challenging. Higher-resolution maps would allow for model building, which would more effectively convey these distinctions.

(8) Including particles number in each class in Fig S8 panel C and D would help in evaluating the quality of classification.

Noted.

(9) A table for cryo-EM statistics should be included.

Thanks, noted.

(10) n values are often provided as a range within legends but it would be better to provide individual values for each dataset. In many figures you can see most of the data points, which is great, but it would be easy to add n values to the plots themselves, perhaps in parentheses above the data points.

While we agree that transparency is essential, adding n-values to each graph would make some figures less clear and potentially harder to interpret in this case. We believe that the dot plots, n-value range, and statistical analysis provide adequate support for our claims.

(11) The way caspase activation of Panx channels is presented in the introduction could be viewed as dismissive or inflammatory for those who have studied that mechanism. We think the caspase activation literature is quite convincing and there is no need to be dismissive when pointing out that there are good reasons to believe that other mechanisms of activation likely exist. We encourage you to revise the introduction accordingly.

Thank you for this comment. Although we intended to support the caspase activation mechanism in our introduction, we understand that the reviewer’s interpretation indicates a need for clarification. We hope the revised introduction removes any perception of dismissiveness.

(12) Why is the patient data in Fig 4F normalized differently than everything else? Once the above issues with mVenus quenching data are clarified, it would be good to be systematic and use the same approach here.

For Fig. 4F, we used a distinct normalization method to account for substantial day-to-day variation in experiments involving body fluids. Notably, we did not apply this normalization to other experimental panels due to their considerably lower day-to-day variation.

(13) What was the rational for using the structure from ref 35 in the docking task?

The docking task utilized the human orthologue with a flipped-up NTD. We believe that this flipped-up conformation is likely the active form that responds to lysolipids. As our functional experiments primarily use the human orthologue for biological relevance, this structure choice is consistent. Our docking data shows that LPC does not dock at this site when using a construct with the downward-flipped NTD.

(14) Perhaps better to refer to double Asn ‘substitutions’ rather than as ‘mutations’ because that makes one think they are Asn in the wt protein.

Done.

(15) From Fig S1, we gather that Panx2 is much larger than Panx1 and 3. If that is the case, its worth noting that to readers somewhere.

We have added the molecular weight of each subtype in the figure legend.

(16) Please provide holding voltages and zero current levels in all figures presenting currents.

We provided holding voltages. However, the zero current levels vary among the examples presented, making direct comparisons difficult. Since we are comparing currents with and without LPC, we believe that indicating zero current levels is unnecessary for this study.

(17) While the authors successfully establish lysophospholipid-gating of Panx1 and Panx2, Panx3 appears unaffected. It may be advisable to be more specific in the title of the article.

We are uncertain whether Panx3 is unaffected by lysophospholipids, as we have not observed activation of this subtype under any tested conditions.